# DNA translocation mechanism of the MCM complex and implications for replication initiation

Martin Meagher [1,3], Leslie B. Epling [1,2,3] & Eric J. Enemark[1]

The DNA translocation activity of the minichromosome maintenance (MCM) complex powers DNA strand separation of the replication forks of eukaryotes and archaea. Here we illustrate an atomic level mechanism for this activity with a crystal structure of an archaeal MCM hexamer bound to single-stranded DNA and nucleotide cofactors. Sequence conservation indicates this rotary mechanism is fully possible for all eukaryotes and archaea. The structure definitively demonstrates the ring orients during translocation with the N-terminal domain leading, indicating that the translocation activity could also provide the physical basis of replication initiation where a double-hexamer idly encircling double-stranded DNA transforms to single-hexamers that encircle only one strand. In this mechanism, each strand binds to the N-terminal tier of one hexamer and the AAA+ tier of the other hexamer such that one ring pulls on the other, aligning equivalent interfaces to enable each hexamer to pull its translocation strand outside of the opposing hexamer.

[1] Department of Structural Biology, St Jude Children's Research Hospital, 262 Danny Thomas Place, Mail Stop 311, Memphis, TN 38105, USA. [2] Present address: Incyte Research Institute, 1801 Augustine Cut-off, Wilmington, DE 19803, USA. [3] These authors contributed equally: Martin Meagher, Leslie B. Epling. Correspondence and requests for materials should be addressed to E.J.E. (email: eric.enemark@stjude.org)

DNA replication is the fundamental process used by all organisms to precisely duplicate the genetic material in preparation for cell division. During DNA replication, a helicase enzyme separates the individual strands of the DNA double-helix so that polymerases can use each strand as a template in the synthesis of new DNA. The helicase uses energy derived from ATP hydrolysis to perform DNA strand separation and also to drive the replication machinery along the DNA to fully replicate it. In eukaryotes, bacteria, archaea, and several double-stranded DNA (dsDNA) viruses, the core of the helicase consists of six proteins arranged in a ring[1–3]. It is widely accepted that these hexameric ring helicases unwind DNA by a strand-exclusion mechanism where the ring encircles one strand and excludes the other[1,3–7]. This topological arrangement propagates DNA separation when the ring moves unidirectionally along the DNA.

In eukaryotes and archaea, six related minichromosome maintenance (MCM) proteins[8] make up the hexameric ring at the core of the replicative helicase. All of these MCM proteins belong to the same structural family defined by an N-terminal domain[9] and an ATPase domain that belongs to the PS2 insert clade[10] of the AAA+ family of ATPases[11]. In eukaryotes, the MCM ring is a heterohexamer of six different related proteins (Mcm2–7) arranged in a specific order[12], while the MCM ring of archaea is generally a homohexamer of six identical proteins. To ensure that each DNA segment is replicated only one time per cell division[13,14], the activities of the Mcm2–7 ring are strictly regulated in a defined sequence of recruitment, loading, and activation. Together with Cdt1, Mcm2–7 rings are recruited to replication origins by the origin recognition complex (ORC)[15–17] and Cdc6 by formation of an ORC-Cdc6-Cdt1-MCM (OCCM) complex[18]. ATP hydrolysis by ORC and Cdc6 drives displacement of Cdt1 and loading of Mcm2–7 rings to the DNA[19] as a double hexamer[20] that encircles dsDNA[21]. This double hexamer can passively slide over dsDNA but does not unwind it[22,23]. At the onset of the S-phase, the activities of cyclin-dependent kinases and the Dbf4-dependent kinase drive three large fundamental changes to activate DNA unwinding[24]: (1) recruitment of Cdc45 and the tetrameric GINS complex to the Mcm2–7 ring to form a larger CMG complex (Cdc45-MCM-GINS[25]); (2) splitting the double hexamer interface to generate two isolated single hexamers[26]; and (3) the ring switches from encircling dsDNA[21] to encircling single-stranded DNA (ssDNA)[5].

Although each of the 11 proteins comprising the CMG is an essential factor for in vitro origin-dependent DNA replication[27], the six proteins of the Mcm2–7 ring by themselves are sufficient to separate base-paired strands in an in vitro unwinding assay[28]. As such, the MCM ring is the minimal complex that can perform generic DNA unwinding, but it is insufficient for origin-dependent DNA unwinding. Similarly, archaeal MCM rings form a larger complex analogous to eukaryotic CMG[29], but the archaeal MCM hexamer by itself is sufficient to unwind DNA in a biochemical assay[30].

An atomic-level description of the mechanism DNA translocation by the MCM ring is fundamental to understanding the replication fork because it provides the central engine. Here we show the 3.15 Å resolution crystal structure of the MCM complex of the archaeal organism *Sulfolobus solfataricus* (*Sso*MCM) bound to ssDNA, ATP analog, and ADP co-factors. The structure indicates that the MCM complex uses a rotary mechanism to translocate DNA. The molecular contacts between *Sso*MCM and ssDNA are fully possible for all MCM proteins, suggesting that the rotary mechanism is shared by all archaeal and eukaryotic MCM helicases. We illustrate how this DNA translocation mechanism can provide the physical basis for unwinding DNA and also for the transformation of a

double-hexamer that encircles dsDNA to single hexamers that each encircle ssDNA.

## Results

**MCM ring structure with multiple nucleotide states and ssDNA.** To elucidate how the MCM ring performs ssDNA translocation, we determined the crystal structure of a nearly full-length *Sso*MCM protein construct in complex with ssDNA and nucleotide co-factors (ADP/ADP-BeF₃; see Methods and Table 1). The crystallized construct (*Sso*MCM-GGSGGS-ΔC) used the C-terminal truncation of a previously reported *Sso*MCM construct that was active in a DNA-unwinding assay[31]. A 9-residue interdomain linker was also shortened to six residues and mutated to the sequence GGSGGS (Fig. 1). The *Sso*MCM-GGSGGS-ΔC construct showed strand displacement in an unwinding assay at similar concentrations to wild-type full-length *Sso*MCM (Fig. 1) and appeared to have stronger cooperativity (Table 2).

The crystal structure asymmetric unit contains one *Sso*MCM hexamer with ssDNA bound in the central channel (Supplementary Fig. 1) and with $Mg^{2+}$/ADP-BeF₃ co-factors bound at three subunit interfaces and with $Mg^{2+}$/ADP at the other three interfaces (Fig. 2). The N-terminal domains and the AAA+ domains of the *Sso*MCM ring form an expected two-tier architecture described previously[20–22,30,32–36]. The N-terminal domain has three subdomains as defined previously[9], consistent with many other MCM structures[9,37,38]. The ATPase domain tier is grossly similar to the ATPase domain tier observed in an earlier ADP-bound crystal structure[39]. The most significant differences within this tier are the presence of DNA and a narrower central

| Table 1 Crystallographic statistics | |
|---|---|
| **Data collection** | |
| Space group | P4₁2₁2 |
| Cell dimensions | |
| $a, b, c$ (Å;) | 186.665, 186.665, 281.489 |
| $\alpha, \beta, \gamma$ (°) | 90, 90, 90 |
| Resolution (Å;) | 50.00–3.15 (3.26–3.15) |
| $R_{sym}$ | 0.119 (1.789) |
| $I / \sigma I$ | 17.6 (0.5) |
| Completeness (%) | 98.0 (89.6) |
| Redundancy | 11.1 (4.5) |
| CC1/2 | (0.245) |
| **Refinement** | |
| Resolution (Å;) | 50.00–3.15 |
| No. of reflections | 84156 |
| $R_{work}/R_{free}$ | 0.1881/0.2413 |
| No. of atoms | |
| Protein | 28,093 |
| DNA | 201 |
| ADP-BeF₃ | 93 |
| ADP | 81 |
| $Mg^{2+}$ | 6 |
| $Zn^{2+}$ | 6 |
| Water | 18 |
| B-factors | |
| Protein | 146.352 |
| DNA | 156.234 |
| ADP-BeF₃ | 102.610 |
| ADP | 122.559 |
| $Mg^{2+}$ | 147.353 |
| $Zn^{2+}$ | 234.377 |
| Water | 80.242 |
| R.m.s. deviations | |
| Bond lengths (Å;) | 0.007 |
| Bond angles (°) | 1.044 |

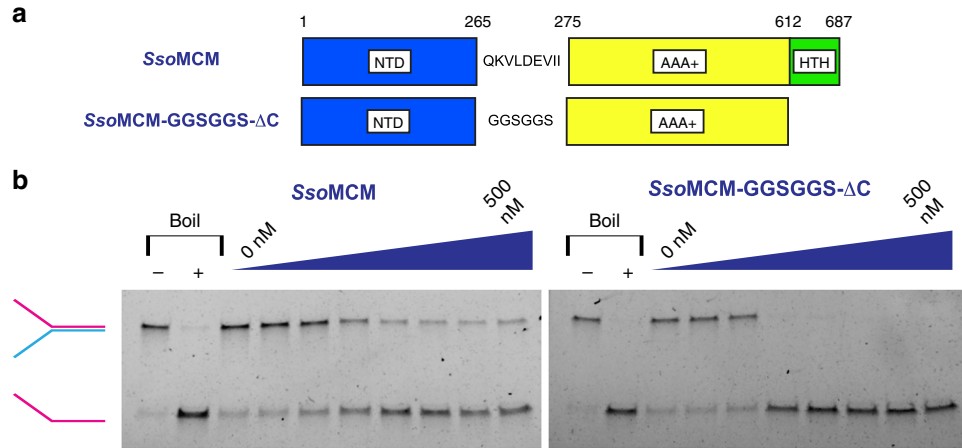

**Fig. 1** The MCM construct of this study, *Sso*MCM-GGSGGS-ΔC, unwinds DNA. **a** The fundamental domain organization of MCM proteins includes an N-terminal domain (NTD), an AAA+ ATPase domain, and a C-terminal helix-turn-helix domain (HTH). The MCM construct of the current structural study consists of *Sso*MCM with the 9-residue (QKVLDEVII) linker between the NTD and the AAA+ domain shortened to six residues and also mutated to the sequence GGSGGS. The construct also uses a previously described deletion of the HTH domain at residue 612 (ref. [31]). **b** The *Sso*MCM-GGSGGS-ΔC construct displaces a labeled strand from a fork DNA substrate at least as well as wild-type full-length *Sso*MCM. Protein concentrations were titrated from 0 to 500 nM (monomer), and reactions were incubated at 69 °C for 60 min. Reactions consisted of 25 mM HEPES (pH = 7.6), 10 mM NaCl, 5 mM Mg (OAc)$_2$, 4 mM ATP, and 3.7 nM fluorescein-labeled DNA substrate

**Table 2 Strand displacement activity parameters**

| Protein | $k_{half}$ | h |
|---|---|---|
| *Sso*MCM | 118.0 ± 19.33 | 1.160 ± 0.2619 |
| *Sso*MCM Y386A | 52.28 ± 17.09 | 0.9462 ± 0.2805 |
| *Sso*MCM K430A | Very large; undetermined | ND |
| *Sso*MCM-GGSGGS-ΔC | 115.2 ± 6.806 | 6.610 ± 2.030 |
| *Sso*MCM-GGSGGS-ΔC Y386A | 132.2 ± 10.55 | 3.310 ± 0.8309 |
| *Sso*MCM-GGSGGS-ΔC T369A | 133.1 ± 28.69 | 0.7792 ± 0.2188 |
| *Sso*MCM-GGSGGS-ΔC K430A | Very large; undetermined | ND |

For each protein, the parameters $K_{half}$ and $h$, along with their standard error, were obtained by fitting two simultaneous Hill equations:
Bottom band: $[MCM]^h (K_{half}{}^h + [MCM]^h)^{-1}$; Top band: $1 − [MCM]^h (K_{half}{}^h + [MCM]^h)^{-1}$, with [MCM] as the monomeric MCM concentration

pore in the DNA-bound structure. The inter-tier orientation of the current structure is highly similar to those of the yeast (*Saccharomyces cerevisiae*, *Sc*) Mcm2–7 double hexamer[20,21] and also *Sc*CMG[36].

**The MCM ring forms a staircase to bind ssDNA.** The MCM ring binds ssDNA in a two-nucleotide per subunit increment with a consistent set of interactions that permute around the ring in a spiral staircase (Fig. 2). The staircased arrangement is analogous to those observed in the crystal structures of other hexameric helicases[4,7,40], and the two-nucleotide per subunit increment matches that observed for DnaB[7]. The 5′-end of the DNA is directed towards the N-terminal domain. Taken together with the known 3′→5′ helicase polarity of MCM[30], the MCM ring orients at the replication fork with the N-terminal domain leading to face the incoming DNA (see DNA translocation). This orientation matches that assigned for *Sc*CMG by an electron microscopy study[41].

The specific MCM:DNA molecular interactions involve two prominent modules of the AAA+ domain that project into the central channel[39]. These form a consistent set of MCM:ssDNA

interactions that permute from one subunit to the next. Phosphate atoms of the ssDNA interact with two positions of the pre-sensor-1-beta-hairpin (ps1β) and with two positions of the helix-2-insert (h2i) (Fig. 2). The MCM ps1β interacts with ssDNA in a nearly identical fashion to the ps1β of papillomavirus E1[4] (Supplementary Fig. 2) with a main-chain amide (A431) interacting with one phosphate of ssDNA and a conserved lysine side-chain (K430) forming an ionic interaction with the ssDNA phosphate located to the immediate 3′-position. The h2i binds ssDNA with two interactions in a similar mode, one derived from the side-chain of T369 and the other from the main-chain amide of V377. The identified interactions are fully possible for all MCM rings because all MCM proteins conserve residues at these positions that are capable of the ionic interactions and hydrogen bonds (Supplementary Fig. 3). The aromatic side-chain of Y386 stacks on the DNA backbone sugar, similar to the interaction of papillomavirus E1 H507 with DNA backbone sugars[4] (Supplementary Fig. 2).

The roles of the residues that contact DNA in the structure were assessed by mutagenesis and a strand displacement assay (Table 2 and Supplementary Fig. 4). The single mutants K430A, T369A, and Y386A were constructed for full-length *Sso*MCM and for *Sso*MCM-GGSGGS-ΔC. The interactions between the K430 side-chains and the phosphate backbone represent the only ionic interactions between MCM and DNA in the structure, and mutation of K430 has previously been shown to ablate DNA-unwinding for full-length *Sso*MCM[42]. Consistent with removing a strong ionic interaction, the K430A mutants exhibited no detectable strand displacement activity for full-length, as shown previously[42], and also for *Sso*MCM-GGSGGS-ΔC K430A. The T369 side-chains form hydrogen bonds with the ssDNA phosphates. The *Sso*MCM-GGSGGS-ΔC T369A mutant exhibited reduced cooperativity of strand displacement, and the full-length *Sso*MCM T369A mutant was badly aggregated and could not be tested in the assay. Taken together, the results for the T369A mutants suggest that T369 may have a role in permitting the subunits to collectively adopt a specific conformation to bind ssDNA during strand displacement. We illustrate below a more general role for the h2i in the vicinity of T369 in interacting with the ps1β of a neighboring subunit that is consistent with such a role. The detectable activity for *Sso*MCM-GGSGGS-ΔC T369A,

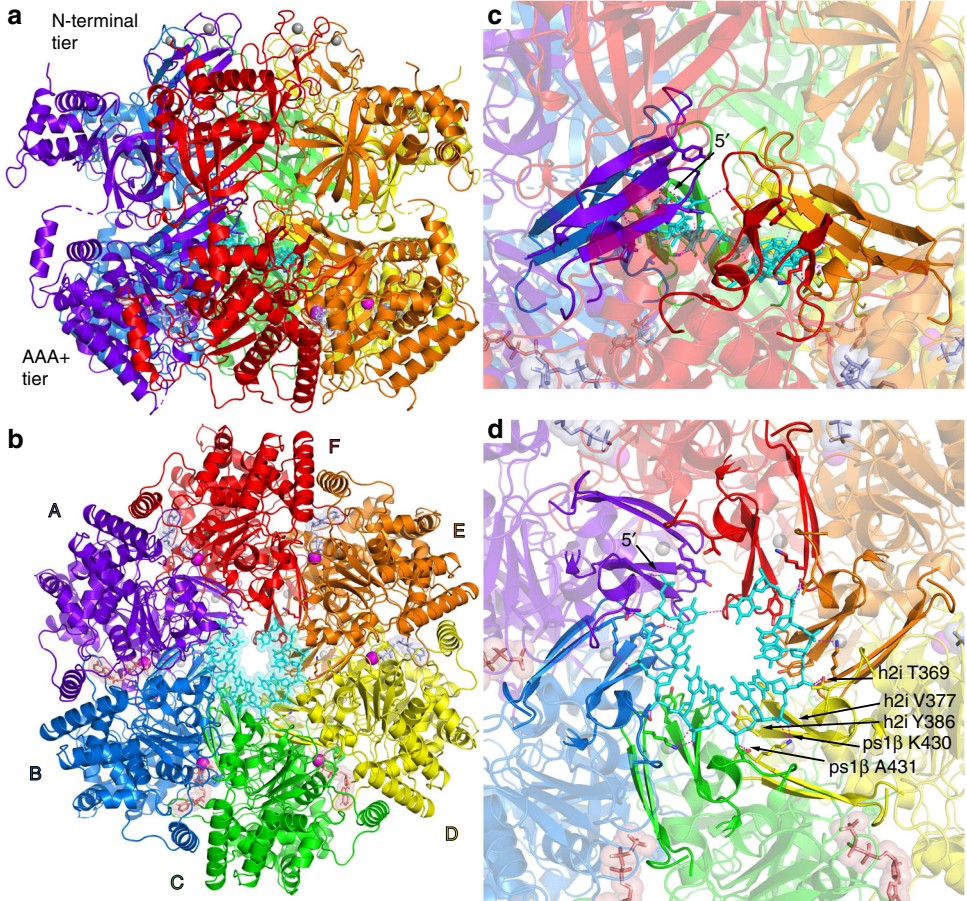

**Fig. 2** 3.15 Å resolution crystal structure of a nearly full-length MCM hexamer bound to ssDNA in a spiral staircase and bound to three ATP-analog molecules and three ADP molecules. Each subunit is uniquely colored with the DNA in cyan. Nucleotide co-factors are in stick and transparent sphere with ADP-BeF$_3$ (ATP analog) in light pink and ADP in lavender. Each subunit projects two hairpins (ps1β and h2i) into the central channel that bind phosphates of the ssDNA with consistent interactions (dashed magenta) that permute around the ring in a spiral staircase. **a** View perpendicular to the central channel with the 5′-end of the ssDNA at the top of the staircase towards the N-terminal tier. **b** View along the central channel from the C-terminal, AAA+ tier side. **c** Zoom-in view of (**a**) with the MCM DNA-binding elements emphasized in opaque. **d** Zoom-in view of (**b**) with the MCM DNA-binding elements emphasized in opaque. See Supplementary Figs. 1–4

in contrast to the abolished activity of the K430A mutants, is consistent with the relatively weaker, non-ionic, hydrogen bonding interactions between T369 and a DNA phosphate. Residue Y386 forms a set of van der Waals interactions with the ssDNA sugars, highly similar to E1 H507 (ref. [4]) (Supplementary Fig. 2). The Y386A mutants did not significantly alter the strand displacement activity compared to *Sso*MCM and *Sso*MCM-GGSGGS-ΔC, consistent with the unperturbed strand displacement activity in mutants of E1 H507 (ref. [43]), and suggesting that Y386 does not have a significant role in strand displacement. We discuss below a potential role for this residue to initiate replication with the conserved aromatic residues on the h2i (including *Sc*Mcm5 F462, *Sc*Mcm2 W589, and *Sc*Mcm6 Y621; see Supplementary Fig. 3) inserting to the DNA minor groove of dsDNA in a directly analogous fashion described for H507 of the E1 hairpin[43].

**Different ATPase states around the ring**. Consistent with other AAA+ complexes, the MCM ring has an ATPase site at each subunit interface[10,12,39], and the ATPase cycle at these sites drives the requisite motion for translocating ssDNA. The bipartite site consists of Walker-A/B and sensor-1 residues of one subunit (referred to as the *cis* subunit) and arginine finger and sensor-2 residues of the neighboring subunit (referred to as the *trans*

subunit). Crystallization of the presented structure relied on the presence of in situ-generated ADP-BeF$_3$. Clear electron density differences in the ATPase sites during the iterative improvement of the crystallographic model (Fig. 3; see Methods) led us to assign ADP-BeF$_3$ molecules at three of the sites, and ADP molecules at the other three sites (see Fig. 2). The structure shows three basic classes of ATPase site that correlate with how tight the interface is and correspond to different positions along the ATPase pathway, analogous to the binding site change rotary mechanism of the F1-ATPase[44,45] and the ATP-type, ADP-type, and apo interfaces described for the E1 helicase[4]. The ATPase sites that contain ADP-BeF$_3$ represent ATP-type, tight interfaces (T) with the site of strongest BeF$_3$ density assigned as the tightest (T*) interface (Fig. 3), the one most like the hydrolysis transition state. The ATP-type site shows continuous density between the BeF$_3$ and the arginine finger, R473, of the *trans* subunit (Supplementary Movie 1). This site also shows a strong interaction between Q423 of the *trans* subunit and the magnesium coordination sphere and also the α-phosphate. Looser ADP-type (D) interfaces lack strong BeF$_3$ difference density (Fig. 3) and do not strongly engage R473, but they retain an interaction with Q423. The apo (E) interface lacks the strong difference density (Fig. 3) and also has lost interaction between the nucleotide and Q423 (Supplementary Movie 1).

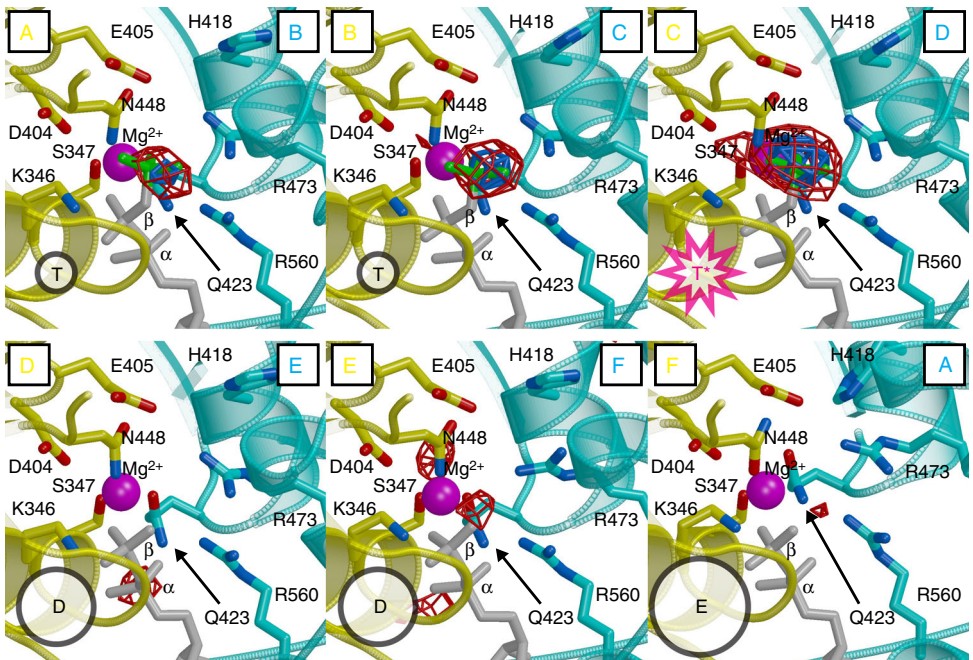

**Fig. 3** Different ATPase states around the ring. Prior to inclusion of BeF₃ in the model, the Fo−Fc difference electron density showed prominent differences at the putative γ-phosphate regions of the ATPase sites, indicating different ATPase states. The electron density is contoured at 4.5-sigma (red) and at 6-sigma (blue). The first three sites (top) showed strong Fo−Fc density (>6-sigma) and were assigned as ATP-type (T). Of these, the interface between subunits C and D showed the strongest density and was assigned as the hydrolysis transition state (T*). The other three sites did not show strong Fo−Fc density in the region and were assigned as loose (D=ADP-type or E=empty). The loss of interaction between Q423 and nucleotide distinguishes empty from ADP-type. The map is drawn on the final model with each subunit labeled according to Fig. 2. The subunit providing the Walker-A (K346, S347), Walker-B (D404, E405), and sensor-1 (N448) residues (*cis* subunit) is in yellow. The subunit providing the arginine finger (R473), sensor-2 (R560), and sensor-3 residues (H418, Q423) is in cyan (*trans* subunit). ADP molecules are in gray stick and BeF₃ molecules are in green stick. See Supplementary Movie 1

## Discussion

The fundamental activity of the helicase ring is to apply a pulling force to one strand to move it through the central channel, an activity termed translocation. A molecular description of DNA translocation has two components—how the MCM ring grips encircled ssDNA and how ATP hydrolysis drives motion of the gripping elements. The spiral staircased coordination of ssDNA and the correlation of each position on the staircase with the status of the subunit's ATPase site indicate that the MCM helicase translocates ssDNA with the ssDNA-binding hairpins descending stepwise and ATP hydrolysis proceeding sequentially around the ring (Fig. 4). As formalized previously[1], a sequential ATP hydrolysis mechanism for a ring helicase bears strong analogy to the rotary catalysis mechanism of F1-ATPase[46] with the catalytically active site (T*) rotating about the ring. Hence, this mechanism is termed rotary. This rotary mechanism encompasses three sequential events that repeat in a cycle: (1) ATP hydrolysis drives the collective MCM:DNA staircase downward (from the N-terminal tier towards the AAA+ tier); (2) release of ATP hydrolysis products; and (3) binding ATP at the empty ATPase site to drive the subunit at the bottom of the staircase to the top. This mechanism is straightforward to illustrate by passing each subunit through all of the identified ATPase and ssDNA-binding states of the structure in sequence (Supplementary Movie 2). A complete cycle binds and hydrolyzes 6 ATP molecules, releases 6 ADP molecules, and translocates 12 nucleotides of ssDNA through the central channel. A rotary translocation mechanism for the MCM ring helicase would unify all ring helicase superfamilies with a common rotary translocation mechanism[4,7,40]. More broadly, other ring-shaped translocases also use a related rotary mechanism, as shown recently for a polypeptide translocase[47].

All MCM hexamers likely translocate ssDNA in this fundamental way based on the conservation of ssDNA-binding atoms in all archaeal and eukaryotic MCM subunits (Supplementary Fig. 3). In order for the h2i of eukaryotic Mcm2–7 to bind ssDNA with these conserved residues, we predict a conformational change is needed for Mcm3, Mcm5, and Mcm2. The PDB models for Mcm2–7 have a horseshoe conformation in the vicinity of the conserved serine/threonine that is immediately N-terminal to the hairpin of the h2i. This region adopts a helical conformation in our ssDNA-bound structure. The horseshoe conformation is present in the DNA-bound and the DNA-free ScMcm2–7 double-hexamer structures (PDB 5BK4 (ref. [21]) and 3JA8 (ref. [20])), and these conformations are well-founded in the corresponding density maps (EMD_9400 (ref. [21]) and EMD_6338 (ref. [20])). In the case of DNA-bound CMG, adjacent subunits are closer together than in the double-hexamer structures, and the horse-shoe conformations of the deposited model (PDB 5U8S[41]) severely clash with the neighboring ps1β (Fig. 5a). These clashes are removed by the helical conformation present in our structure. To illustrate the compatibility, a model for CMG:DNA was constructed by converting the residues of our *Sso*MCM-GGSGGS-ΔC:ssDNA structure to the Mcm2–7 sequences and aligning the resulting subunit domain:ssDNA structures to those of CMG:DNA (Fig. 5b). The resulting model maintains all the MCM:DNA interactions of the *Sso*MCM-GGSGGS-ΔC:ssDNA structure (which is inherent in the construction method), and provides a reasonable fit to the map (EMD_8518 (ref. [41])). In the helical conformation, two parallel-β interactions are established

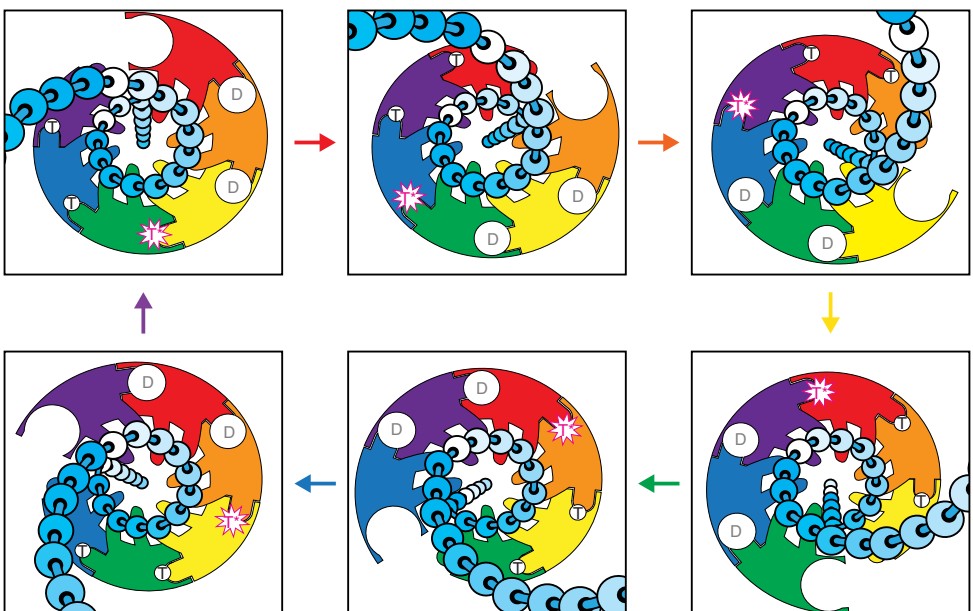

**Fig. 4** Cartoon representation of the DNA translocation mechanism viewed from the C-terminal side (beneath the staircase). The subunits are colored as in Fig. 2 with individual DNA nucleotides represented as beads on a string in a variable gradient shade of cyan. For each transition, the arrow color indicates the subunit moving from the bottom of the staircase to the top. Each subunit passes sequentially through the full complement of DNA-binding and ATPase states with its two DNA-binding hairpins escorting two nucleotides of ssDNA through the central channel (see Supplementary Movie 2)

between the h2i of one subunit and the ps1β of the neighboring subunit. The first interaction is between the amide of the conserved serine/threonine of the h2i (T369 in SsoMCM) and the carbonyl one residue N-terminal of the conserved lysine on the neighboring ps1β (A429 in SsoMCM). The second interaction, characteristic of a parallel β-sheet, involves the amide two amino acids N-terminal to the conserved serine/threonine of the h2i (G367 in SsoMCM) and the carbonyl of the same residue of the first interaction on the ps1β (A429 in SsoMCM). These interactions stabilize the different step heights of the staircase, and thus represent staircasing interactions analogous to those described for the E1 helicase[4]. The transition of the Mcm5/2 interface from unstaircased with a horseshoe h2i conformation to staircased with a helical h2i conformation is illustrated in Supplementary Movie 3. We suggest that the capacity of the h2i and ps1β to adopt a parallel-β staircased structure may correlate with the cooperativity observed in the strand displacement assay (Table 2 and Supplementary Fig. 4).

Although each ATPase site generally participates in driving translocation in a rotary mechanism, the mutation of one ATPase site might not disable translocation altogether. Mutation of one ATPase site is analogous to removing one spark plug from a multi-cylinder engine. Generally, removal of one spark plug will not disable the entire engine. The comparable role of each ATPase site in the rotary mechanism (Fig. 4) might lead to the prediction that all single ATPase site mutants should behave equivalently. However, individual lysine-to-alanine Walker-A mutants of CMG have shown large differences in strand displacement activity[24]. Mcm3/5 and Mcm5/2 ATPase site mutants essentially abolish strand displacement while other ATPase site mutants have little effect[24]. Notably, the subunits comprising the susceptible ATPase sites (Mcm3/5; Mcm5/2) are the precise subunits where the h2i must remodel from horseshoe to helix to bind ssDNA (Fig. 5). This remodeling is associated with bringing adjacent subunits closer together (see Supplementary Movie 3)— which is precisely what occurs upon ATP-binding. We therefore suggest that the underlying reason that the Walker-A mutants of

Mcm3 and Mcm5 are defective in unwinding DNA is that they fail to tighten their intersubunit interfaces sufficiently to drive the h2i of Mcm3, 5, and 2 from horseshoe to helix. Without such remodeling, the h2i of these subunits would be in conformations more similar to those of the inert double-hexamer than the unwinding-competent form depicted in our structure. Consistently, the subunits of the non-susceptible ATPase sites, Mcm6, Mcm4, and Mcm7, do not require remodeling of the h2i because each adopts the helix conformation appropriate for binding ssDNA within both double-hexamer structures[20,21]. Since the h2i of Mcm6, 4, and 7 appear to intrinsically adopt a conformation appropriate for ssDNA-binding, we suggest that mutation of their corresponding ATPase sites constitutes removing one spark plug from a multi-cylinder engine in the above analogy, and therefore the Walker-A mutants involving Mcm6, Mcm4, and Mcm7 are able to retain strand displacement activity.

The MCM ring unwinds dsDNA through a strand-exclusion mechanism such that the ring encircles the translocating strand while excluding its complementary strand[5]. When the translocating strand is pulled through the central channel, the topological barrier of the ring peels off the complementary strand. Strand-exclusion unwinding minimally encompasses three forces acting on the DNA: tension on the encircled strand directed along the channel axis generated by translocation by the ring; curvature of this strand around the dsDNA helical axis arising from dsDNA base-pairing; and a normal force of the ring against the duplex DNA. The situation is highly analogous to the three forces of a pulley system: rope tension from a tethered mass, rope curvature due to a pulley wheel, and a normal force that maintains a fixed pulley axis. The B-DNA cylinder is conceptually similar to a pulley wheel except that the forces of curvature follow a helical path rather than lying in a plane.

The force of tension on the encircled strand requires that its direction does not change without the application of additional forces. Consequently, if no additional forces are invoked, the direction of ssDNA translocation will coincide with the trajectory of the strand in its final helical segment in the dsDNA duplex

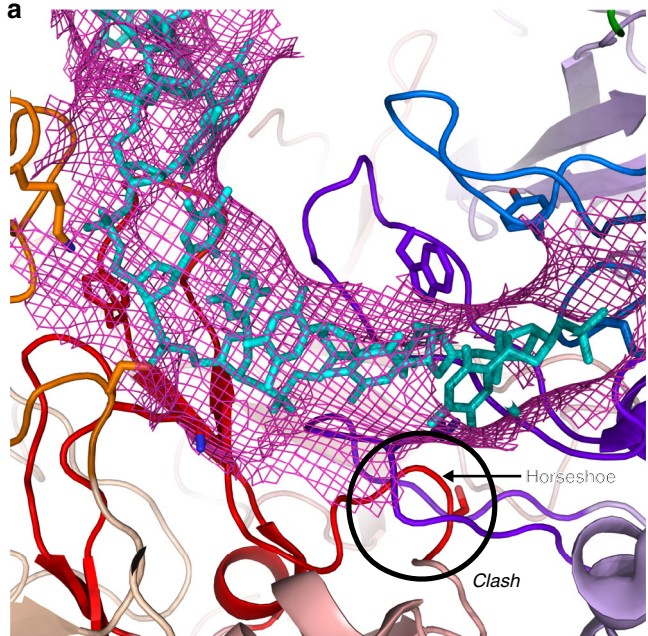

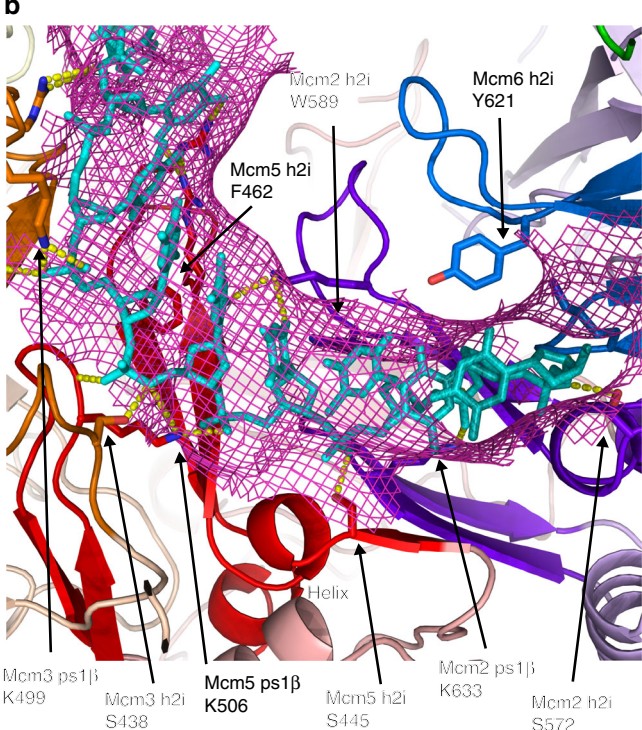

**Fig. 5** An intersubunit clash when binding ssDNA is removed when a horseshoe h2i conformation is converted to a helical conformation. **a** The CMG:DNA model (PDB 5U8S[41] has a horseshoe conformation in the h2i of Mcm3, Mcm5, and Mcm2 that severely clashes with the ps1β of the neighboring subunit. **b** This clash is removed when the horseshoe portion is modified to helix. Mcm3 is in orange, Mcm5 in red, and Mcm2 in purple. The ps1β and h2i are emphasized in bold color with the rest of the protein in light color. The model in panel **b** was generated from our MCM:ssDNA structure by Chainsaw[66] followed by brief geometry minimization with phenix.geometry_minimization[59]. The map of CMG:DNA (EMD_8518 (ref. [41])) is drawn in the vicinity of the DNA for both panels

(Fig. 6, Supplementary Movie 4). This relationship is equivalent to that of a pulley system where the direction of a rope as it exits contact with the wheel coincides with the direction of tension. Using this arrangement, the translocation of two nucleotides of ssDNA in one step of the cycle breaks the base-pairing interactions for the last two base-paired nucleotides and also rotates the remaining intact dsDNA about its helical axis by the twist of two nucleotides (72° for ideal B-form DNA), similar to the rotation of a pulley wheel. This rotation directly quantifies the extent of unwinding of the two strands with the 72° rotation illustrated in Fig. 6 corresponding to a net unwinding of 72°.

The orientation of the MCM ring during DNA translocation with the N-terminal domain leading implies that the constituent hexamers of the double-hexamer must pass each other in order to establish two bidirectional replication forks. An analogous pass by transformation has been described for the double-hexamer of papillomavirus E1 (refs. [4,48]) as well as for eukaryotic CMG[41]. Given this orientation, we suggest that the transformation of an idle double hexamer encircling both DNA strands to two independent single hexamers that encircle and translocate one strand will proceed with each MCM hexamer pulling its translocation strand outside of the opposing hexamer. We provide a hypothesis for the mechanism of this transformation (see below) that proceeds via two discrete steps of DNA unwinding, consistent with activation of eukaryotic Mcm2–7 (ref. [49]). Activation of eukaryotic Mcm2–7 depends on recruitment of Cdc45, GINS, and Mcm10 (ref. [49]), which are discussed in the next section.

For the constituent hexamers of the double-hexamer to begin pulling the DNA strands, each ring must first establish a grip on its translocation strand as in the staircase of Fig. 2. When one strand binds to the AAA+ domain in this polarity (counter-clockwise 5′ to 3′ when viewed from the C-terminal side), its complementary strand is ideally positioned to bind in the plane of the N-terminal domain tier at the MCM single-stranded binding motif (MSSB) with the binding mode and polarity shown previously (clockwise 5′ to 3′ when viewed from the C-terminal side)[38] (Fig. 7). The polarities of these strands, and the positions immediately above and below the h2i[38,39], indicate that the h2i projects into the dsDNA minor groove in establishing these interactions. Intriguingly, an aromatic residue conserved on the h2i of archaeal MCMs and three adjacent subunits of eukaryotic MCMs (Mcm5, Mcm2, Mcm6) (Supplementary Fig. 3 and Fig. 5) sits in a very similar position to the aromatic residue (H507) of the hairpin of the papillomavirus E1 helicase (Supplementary Fig. 2)[4], which projects into the minor groove of dsDNA and plays essential role in replication initiation[43]. The aromatic residue of the MCM h2i may function similarly to E1 H507 to enable replication initiation. A model for the SsoMCM:DNA species with h2i inserted to the minor groove is depicted in Figs. 7 and 8, where the orientation of the starting staircase with respect to the overall dyad axis is based on the dyad of the Mcm2–7 double hexamer[50] and a premise that the initial staircase is set by closure of the Mcm5/2 ATPase site according to the slow step of DNA-encircling described by Bochman and Schwacha[34]. The binding of ATP to this ATPase site will drive the Mcm5 hairpins to the top of the staircase because the transition of an ATPase site from empty (E) to ATP-bound (T) drives the associated hairpins to the top of the staircase (see Fig. 4 and Supplementary Movie 2).

In order for each hexamer to pull its translocation strand outside of the opposing hexamer through equivalent interfaces, the two interfaces would need to align to face each other. This orientation permits the strands to simultaneously be pulled through the two matched interfaces, similar to dental floss. Such alignments can be generated by either twisting the two hexamers with respect to each other or by sliding them (Supplementary Movie 5). In the case of twisting, one direction will partially

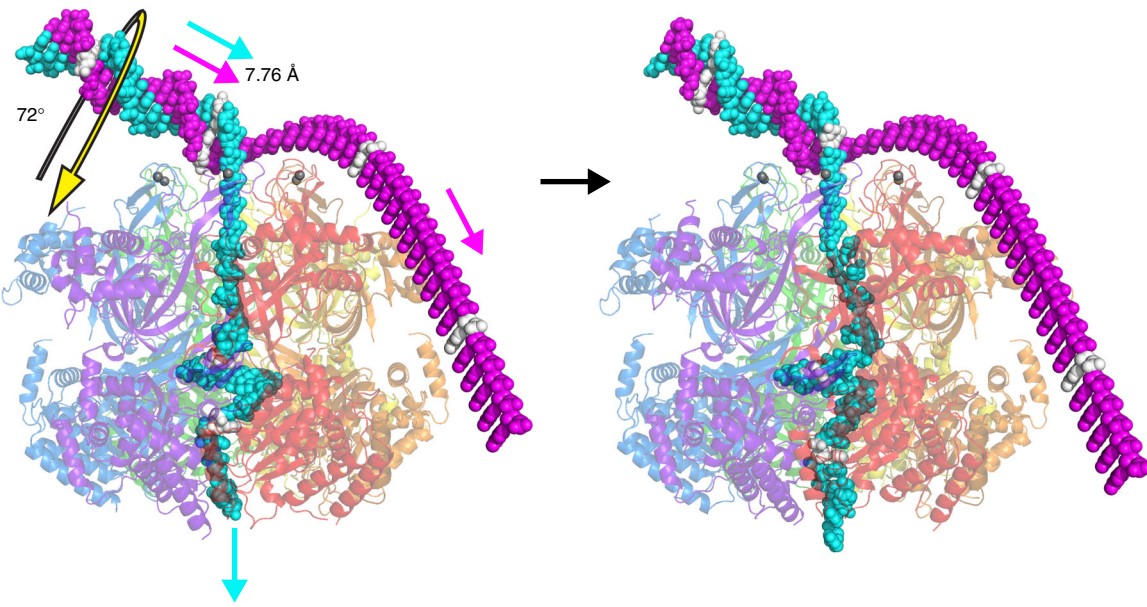

**Fig. 6** Proposed DNA unwinding mechanism by the MCM complex. The figure depicts one step of the translocation cycle (see also Supplementary Movie 4). The hexameric ring pulls two nucleotides of the cyan strand through the central pore to provide two base-pairs of unwinding (72° rotation and 7.76 Å translation for idealized B-form DNA). Each subunit is uniquely colored in transparent cartoon, and the two DNA strands are in cyan and magenta spheres. Every 12th DNA nucleotide is colored white to help show strand motion. In this orientation, the ring exerts a downward force on the cyan strand (translocation) and an upward normal force on the magenta strand (exclusion)

unwind the DNA, and the other direction will wind it tighter. An interhexamer slide does not rotate the strands and therefore provides zero net change to DNA winding. We consider an interhexamer twist in the unwinding direction as the most likely means to align equivalent interfaces because the associated unwinding is consistent with the first of two unwinding steps towards activation[49]. Unlike a slide, a twist does not require all the interhexamer interfaces to be broken uniformly. Intriguingly, a hinge rotation topologically captures the strands at a specific interface (see below). Two hinge unwinding scenarios are possible depending on whether the hinge is located at the major or minor groove side of the dsDNA. We illustrate the hinge at the major groove side because this geometrically fits an initial staircase that emulates placement of the Mcm5 hairpins at the top (Fig. 8).

The dual attachment of each strand provides a straightforward physical basis for the two hexamers to achieve such a twist where the AAA+ domain of one hexamer pulls on the N-terminal domain of the opposing hexamer. The strand bound at the MSSB is approximately in the plane of the N-terminal tier[38], and pulling on this strand can straightforwardly twist the hexamer. An advantage of this model is that one interhexamer interface persists through the transformation, functionally behaving as a hinge. This hinge forms a topological barrier to the DNA strands that entraps the strands at an interface with the adjacent subunit. The facile nature of a hinge rotation generated by tension applied to the strands is illustrated by the physical model of Supplementary Movie 6 (see also Supplementary Fig. 5 for details), including how the strands are systematically captured between the orange and yellow subunits during rotation about the orange subunit hinge.

Once identical interfaces of the hexamers align to face each other, each hexamer would be appropriately positioned to pull its own translocating strand outside of the other hexamer (Supplementary Movie 7), where it would become the excluded strand. This process involves multiple ATP hydrolysis events at each hexamer that pull each strand progressively tighter. This

unwinds the strands at the center of the double-hexamer while concurrently re-winding them beyond the DNA-binding hairpins to generate a net unwinding of zero degrees (Fig. 8). The re-winding aspect is analogous to winding a spring, and we suggest that unspringing this portion of DNA creates the final exit of the excluded strand from the ring (Fig. 8). At the specific timing when the ATPase site of the exit interface is in the E state (Fig. 4), a large quantity of unwinding is produced in a single burst (Fig. 8), consistent with the second unwinding step towards activation[49]. Although an empty ATPase site is consistent with the open interface needed for the DNA to pass through, the size of the opening depicted for translocation and unwinding (see Supplementary Movie 4) would likely need to be enlarged further to allow such passage. Following final exit, the two MCM:DNA species are both in the appropriate topological form for DNA unwinding by strand-exclusion.

We performed an analogous analysis with Mcm2–7 to include the factors Cdc45 and GINS in our initiation hypothesis at stages of unwinding according to the two-step activation[49] to illustrate that the proposed transformation is physically reasonable (Supplementary Movie 8 and Supplementary Fig. 6). Prior to the dual-unwinding steps, nucleotides are dissipated from the ATPase sites of the double-hexamer, and ATP hydrolysis does not occur until Mcm10 has been recruited to the complex[49]. Hence, the initial steps are energetically driven by the binding energy associated with factor recruitment and/or ATP-binding, and not by ATP hydrolysis. As above, the first step is for each subunit to bind the translocating strand in a staircase as in Fig. 2. This configuration requires remodeling of the structure of the pre-activation Mcm2–7:dsDNA double hexamer[21] such that the ps1β hairpins switch from binding the lagging strand to binding the leading strand. This conversion could be achieved by rotating the double-hexamer relative to the dsDNA or by sliding it along the dsDNA. We illustrate this conversion with a slide (Supplementary Movie 8) because this slide provides the most straightforward

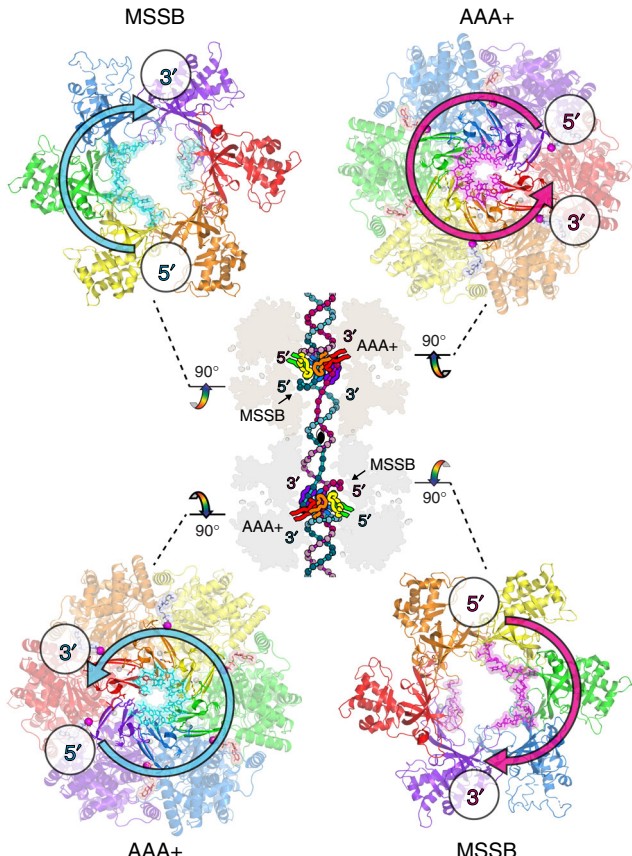

**Fig. 7** ssDNA-binding modes observed for the MCM tiers indicate the h2i projects into the minor groove to establish a translocation configuration for the double-hexamer encircling dsDNA. At each hexamer, the AAA+ domain binds one strand polarity (this work) while the N-terminal domain MSSB binds the opposite polarity— predominantly at Mcm6, 4, and 7 in the case of Mcm2–7— as demonstrated previously (PDB 4POG[38]). Each strand is tethered to the AAA+ domain of one hexamer and the N-terminal domain of the opposing hexamer, indicating that a pulling force at one AAA+ tier would be applied to the N-terminal tier of the other hexamer (and vice versa)

transform and also because the Mcm2–7 double-hexamer is known to slide on dsDNA[22,23].

Following this slide, the h2i can insert to the minor groove such that the leading strand binds to the AAA+ domain hairpins in a staircase, and the lagging strand binds to the MSSB, predominantly at Mcm6, Mcm4, and Mcm7 (ref. [38]). Insertion of the h2i to the minor groove generates zero degrees of unwinding, and is therefore consistent with the unwinding upon recruitment of Cdc45 (ref. [49]). On this basis, we speculate that when Cdc45 binds to the Mcm5/2 interface[35,51], it drives the Mcm5/2 ATPase site to a T state, ensuring the Mcm5 hairpins occupy the top of the staircase. By bringing the Mcm2 ps1β into close proximity with the Mcm5 h2i, it also converts the Mcm5 h2i from a horseshoe conformation to a helical conformation (see Fig. 5 and Supplementary Movie 3), allowing both subunits to interact with ssDNA as in our structure. Importantly, binding of Cdc45 to Mcm2–7 does not appear to sterically interfere with any of the downstream steps in the proposed transformation towards strand exclusion (Supplementary Movie 8).

GINS recruitment to Mcm2–7 is associated with a limited degree of unwinding[49]. Consequently, GINS recruitment could serve as a trigger for a hinge rotational remodeling of the double-hexamer to align equivalent interfaces to face each other (Supplementary Movie 8; see Supplementary Movie 5). Binding of GINS to the MCM ring does not appear to sterically interfere with the suggested hinge rotation or any of the downstream steps towards strand exclusion (Supplementary Movie 8). We speculate that binding of GINS to the Mcm3 and Mcm5 subunits[36] drives the Mcm3/5 ATPase site to a T state, causing the Mcm3 hairpins to move to the top position of the staircase and also converting the Mcm3 h2i from horseshoe conformation to helical conformation. This species, with Mcm3 hairpins at the top of the staircase, would appear very similar to the yeast CMG:DNA structure (PDB 5U8S[41]) except that the h2i would be remodeled from horseshoe to helix as in Fig. 5b. As no ATP hydrolysis is observed prior to Mcm10 recruitment[49] (see below), the energetic driving force for this hinge remodeling would derive from the binding energy associated with GINS recruitment.

Recruitment of Mcm10 is associated with a burst of unwinding[49] that is qualitatively consistent with the unwinding burst described in the unspringing step that provides final strand exit of our proposed transformation. Mcm10 recruitment also initiates the ATP hydrolysis activity. The ATP hydrolysis activity following Mcm10 recruitment could provide the energetic driving force for winding the spring and thus the ultimate unspringing to follow. On this basis, we speculate that Mcm10 would be recruited following completion of step 2 in Fig. 8. Mcm10 recruitment could also serve to trigger the unspringing exit, perhaps by further enlarging the E ATPase site interface to facilitate the DNA passing through.

Many of the features presented in our MCM activation mechanism are generally feasible regardless of the subunit that begins at the top of the staircase. We strongly favor the Mcm5 hairpins moving to the top to begin the process. On this basis, we assign the Mcm3:Mcm3′ interhexamer interface of the double-hexamer as the hinge due to its proximity to Mcm5 and the limited slack of DNA available. An Mcm3:Mcm3′ hinge indicates that an unwinding twist would align the Mcm7/3 interfaces to face each other, and we therefore predict that the excluded strand of each hexamer will exit the ring through the Mcm7/3 interface to establish strand exclusion.

## Methods

**Cloning and mutagenesis**. An N-terminal His₆-SUMO-SsoMCM expression construct was prepared to generate nearly full-length SsoMCM with a shortened linker between the N-terminal domain and the AAA+ domain (pEE078.1). The original SUMO vector was the generous gift of Dr. Christopher D. Lima[52] under MTA. An existing His₆-SUMO-tagged-fusion protein expression construct for full-length SsoMCM in a pRSFduet (Novagen) plasmid (pEE045.1 (ref. [39])) was modified to encode a protein with a shortened linker between the N-terminal domain and the AAA+ domain (amino acids 266–274 replaced by the six residue sequence GGSGGS) by circular ligation of a plasmid-length phosphorylated PCR product. A single-colony plasmid was isolated, sequence verified, and further modified to encode a C-terminal truncation after amino acid 612 (a truncation described in ref. [31]) by circular ligation of a plasmid-length phosphorylated PCR product. A single-colony plasmid was isolated and sequence verified (pEE078.1). DNA sequencing was performed by the Hartwell Center DNA Sequencing Facility (St. Jude Children's Research Hospital). Point mutants (T369A, Y386A, or K430A) were generated similarly by site-directed mutagenesis of pEE045.1 or pEE078.1.

**Expression and purification**. Expression plasmid was transformed into BL21 (DE3)-RIPL (Agilent Technologies) chemically competent cells and grown overnight in a 100 mL starter culture containing 30 mg/L kanamycin. The starter culture was distributed among 6 L of LB media containing 0.4% glucose and 30 mg/L kanamycin and grown to an OD of 0.3 at 37 °C when the temperature was lowered to 18 °C. When the OD had reached 0.7, expression was induced by 0.4 mM IPTG, and the cells were grown for 16 h at 18 °C and harvested by centrifugation. The cells were resuspended and lysed with a microfluidizer, and the soluble fraction was isolated by centrifugation. The sodium chloride concentration was raised to 1 M, and polyethylenimine was added to 0.3% final concentration, and the supernatant was isolated after centrifugation. Ammonium sulfate was added to 70% saturation, and the precipitate was isolated by centrifugation, resuspended, and purified by Ni-NTA (Qiagen) chromatography. The elution was further purified by size-exclusion

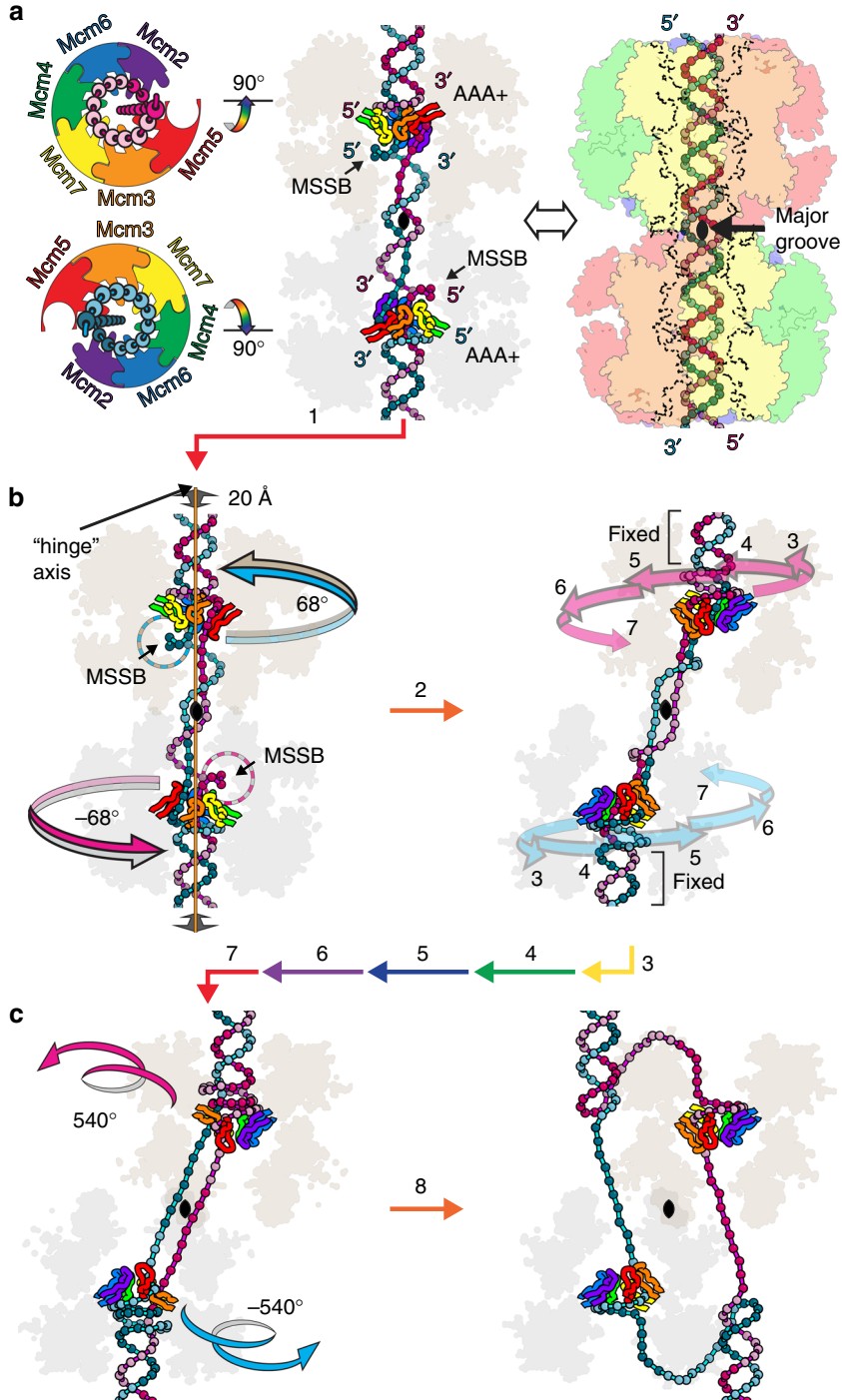

**Fig. 8** Proposed MCM:DNA aspects of replication initiation. The dyad axes of the double-hexamer and the DNA are colinear, and the color of each step arrow indicates the subunit that ascends to the top of the staircase, as in Fig. 4. The hexamers are in shaded outline with the h2i hairpins in colored coil, and the DNA nucleotides are shown as beads on a string. See Supplementary Fig. 6 and Supplementary Movie 7. **a** First, the MCM DNA-binding modules engage binding with the AAA+ domain binding the translocation strand as in Fig. 2 and with the MSSB binding the other strand[38]. The double-hexamer: DNA orientation derives from the established interhexamer orientation[20,21,50] and the premise the Mcm2–7 AAA+ domain initially drives Mcm5 to the top of the staircase. **b** Each hexamer pulls one strand that is tethered to the MSSB of the underlined(opposing) hexamer to drive a hinge rotation about the orange interhexamer interface to align the interfaces of the yellow and orange subunits. Subsequent translocation (steps 3–7) unwinds DNA at the center while winding the translocating strand around the opposing strand, analogous to winding a spring. **c** The region unsprings when the ATPase site between the orange and yellow subunits is empty, completing the transformation from encircling dsDNA to encircling ssDNA

chromatography, and Ulp1 protease was added, and the sample was placed overnight in a dialysis chamber to simultaneously exchange the solution buffer (20 mM Tris pH 8.3; 100 mM NaCl; 5 mM β-ME) and also remove the SUMO tag (the Ulp1 protease plasmid was the generous gift of Dr. Christopher D. Lima[52] under MTA). Cleavage of the SUMO tag was verified by SDS-page, and the MCM protein was

purified by anion exchange followed by gel-filtration chromatography. The proteins elute in a single homogeneous peak at a volume consistent with a hexameric complex. SDS-PAGE showed a single band consistent with the anticipated protein (SsoMCM-GGSGGS-ΔC or SsoMCM) based on proteomic analysis (St. Jude Proteomics center).

**Strand displacement assay**. The unwinding experiment is highly similar to that used previously[39]. The assay uses a Y-shaped DNA substrate with a 55-mer double-stranded region, a 50-mer poly-dT 3′-arm, and a 30-mer 5′-arm with a fluorescein label at the 5′-end. The substrate was prepared by annealing a 5′-fluorescein-labeled oligonucleotide (5′-TTGAACCACCCCCTTGTTAAATCACTTCTACTT GCATGCCTGCAGGTCGACTCTAGAGGATCCCCGGGTACCCGAGCTCGAAT TCG-3′) with an unlabeled oligonucleotide (5′-cgaattcgagctcggtacccggggatcctctaga gtcgacctgcaggcatgcaagTTTTTTTTTTTTTTTTTTTTTTTTTTTTTTTTTTTTTTTT TTTTTTTTTTT-3′, Sigma-Aldrich, St. Louis, MO). The 85-mer oligonucleotide was identical to a previously published substrate that had been annealed to M13 plasmid ssDNA to study DNA-unwinding by SsoMCM[53]. Helicase activity assays were prepared in 20 μL reaction mixtures with 25 mM HEPES (pH = 7.6), 10 mM NaCl, 5 mM Mg(OAc)₂, 4 mM ATP, and 3.7 nM fluorescein-labeled DNA substrate. The protein concentration was titrated from 0 to 500 nM (monomer), and reactions were incubated at 69 °C for 60 min. Prior to the addition of ATP to initiate unwinding, reactions were incubated at 69 °C for 5 min to allow for thermal equilibration. For all samples, reactions were stopped by the addition of 5 μL of loading buffer containing 40% (v/v) glycerol, 5% (w/v) sodium dodecyl sulfate (SDS), and 50 mM ethylenediaminetetraacetic acid (EDTA), and a 20 μL aliquot was loaded on a 4–20% 1× TBE gradient PAGE gel (Invitrogen) and run at 250 V for 45 min. Gel imaging was performed with a GE Amersham Imager 600 using a 10-min exposure time with a blue light source (460 nm) and Cy2 filter.

The intensities of the two bands were integrated for each of the 10 lanes of the assay gel by MultiGauge. Each intensity was divided by the sum of the mean intensity of the top band plus the mean intensity of the bottom band to provide an overall mean total intensity of 1.0 in each line. The intensities of the two bands for each titration point (lanes 3–10) were fit to two simultaneous equations by GraphPad Prism to related Hill equations. The bottom band was fit to: $[MCM]^h (K_{half}{}^h + [MCM]^h)^{-1}$, and the top band was fit to $1 - [MCM]^h (K_{half}{}^h + [MCM]^h)^{-1}$, with $[MCM]$ corresponding to the concentration of MCM monomer. The overall fit of two parameters to two intensities at eight different MCM concentrations provides an 8:1 data/parameter ratio. The parameters were determined from two replicates and reported in Table 2 along with their standard error. Uncropped versions of each gel and also parameter fitting are illustrated in Supplementary Fig. 4.

**Crystallization and data collection**. Long rod-shaped crystals (0.030 × 0.030 × 1.0 mm) of SsoMCM-GGSGGS-ΔC in complex with a 12-mer oligonucleotide, magnesium, ADP, and beryllium fluoride grew at 18 °C in a hanging drop containing 1 μL of protein solution and 1 μL of well solution (50 mM MES, pH 6.0, 100 mM NaCl, 20 mM MgCl₂, 9% PEG 8000). Crystals were harvested with elliptical litholoops (Molecular Dimensions) to align the long dimension of the crystal approximately parallel to the spindle axis to maximize the attainable resolution limit (see below) and to reduce background X-ray scatter. Harvested crystals were briefly immersed in a solution of 33% ethylene glycol in well solution and flash frozen in liquid nitrogen. Data were collected at SER-CAT beamline 22-ID at the Advanced Photon Source at Argonne National Lab. Trial crystals empirically indicated that radiation decay became significantly detrimental to diffraction quality after 250 s of X-ray exposure and that the resolution limit improved with increased exposure times up to an exposure rate of 10 s per degree of rotation (the maximum rate tested). Based on these factors and the high symmetry of the space group (P4₁2₁2), data were collected from six different positions evenly spaced on the long dimension of a single crystal. Data were collected in 25° wedges at an exposure rate of 10 s per degree with consecutive wedges overlapped by 5°. The specific absolute angles of the 6 wedges were: 0–25, 21–45, 41–65, 61–85, 81–105, and 101–125. All individual diffraction images were collected at 100 K with an oscillation width of 0.25°. The data from the six different crystal positions were integrated and scaled with the HKL-2000 package to 3.15 Å resolution with statistics summarized in Table 1. The data exhibit a signal-to-noise greater than 1.0 up to 3.3 Å resolution. Beyond 3.3 Å, the signal-to-noise decreases below 1.0. We include all data to 3.15 Å resolution in the crystallographic refinement because the 3.26–3.15 Å resolution shell is where the $CC_{1/2}$ value falls to 0.245, a criteria that is consistent with recent discussions on the benefits of including weak diffraction data upon the overall quality of crystallographic maps and models[54] and also an analysis indicating that the high resolution range where little information remains occurs where the $CC_{1/2}$ value has dropped to a range of 0.2–0.4 (ref. [55]).

**Structure solution and refinement**. Initial phases were determined by molecular replacement by the program Phaser[56] that placed six copies of a monomer of the N-terminal domain of SsoMCM (PDB 2VL6 (ref. [37])) and six copies of a monomer of the AAA+ domain of Sso-PfMCM (PDB 4R7Y[39]). Following this placement, difference maps revealed strong electron density within the hexameric channel, but this density was not modeled at this stage. The protein model was iteratively refined and manually improved until advancement ceased. Next, the strong difference electron density at each ATPase site was modeled as magnesium-ADP, and the model was iteratively refined and manually improved. At this stage, difference electron density inside the channel was observed at the 7-sigma level (see Supplementary Fig. 1), and it was assigned as ssDNA. The electron density indicated that the ssDNA polarity was with the 5′-end towards the N-terminal domain side and the 3′-end towards the AAA+ domain side. At nearly all nucleotide links, the continuity of the density persisted to a higher sigma level in one direction versus

the other, indicating the 3′-linkage direction due to the decreased conformational flexibility of the 3′-linker relative to the 5′-linker. This correlation was present when the density was examined at each side of the phosphates and also when examined at each side of the sugars. To further confirm the correctness of the assigned polarity, opposite polarity models were independently constructed and refined, and the resulting difference electron density was evaluated, a protocol highly similar to that described previously[4]. This analysis reinforced the correctness of the assigned polarity with the 5′-end at the N-terminal domain side and the 3′-end at the AAA+ domain side. The assigned polarity model was straightforward to build and fit the initial electron density well throughout. Following refinement, no glaring difference electron density was observed. In contrast, the opposite polarity model was awkward to build, did not provide a compelling overall fit to the initial density, and the difference electron density indicated numerous problems that could all be solved by a model polarity with the 5′-end facing the N-terminal side.

The model was again refined and improved until advancement ceased, and the difference electron density at the position for an ATP gamma phosphate was then evaluated (Fig. 3). At this stage, the top three difference peaks were at the ATPase site interfaces of the A/B; the B/C; and the C/D subunits in the region expected for a gamma phosphate of ATP. These were assigned as BeF₃, and their respective interfaces were assigned as tight (T) ATP-type interfaces. Of these, the strongest density was at the C/D interface, and this was assigned as the tightest (T*) interface—the interface that most resembles the hydrolysis transition state. The model was refined at various stages with CNS[57,58], phenix[59], and refmac5 (ref. [60]). The final refinement was carried out with phenix and included 25 TLS groups: four for each subunit (N-terminal domain subdomains A, B, and C and the AAA+ domain) and one for the ssDNA. Model-building was performed with Coot[61]. A Feature Enhanced Map (FEM)[62] calculated based on the final refinement is shown for the ssDNA region (Supplementary Fig. 1b) and for the ATPase sites (Supplementary Movie 1). Figures were prepared with the programs PyMol[63], Bobscript[64], Raster3D[65], and Adobe Illustrator.

**Movie generation and analysis**. Supplementary Movies 1, 2, 4, 7, and 8 illustrate, with increasing layers of complexity, progression of the MCM:ssDNA complex through six successive states in a cycle. Although the crystal structure has a single MCM hexamer bound to ssDNA, the details of each subunit and also each ATPase site differ. Altogether, these define six distinct sequential states in a cycle, illustrated in Fig. 4 and Supplementary Movie 2. During the cycle, each subunit passes through all six distinct conformations observed in the crystal structure. Specialized movie generation details are provided in the Supplementary Methods.

**Reporting Summary**. Further information on research design is available in the Nature Research Reporting Summary linked to this article.

## Data availability
The crystallographic coordinates have been deposited at the PDB accession number 6MII. All data are available from the authors upon reasonable request.

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

## Acknowledgements

We thank Jessie Newman and the St. Jude Department of Biomedical Communications for assistance with Supplementary Movie 6. Data were collected at Southeast Regional Collaborative Access Team (SER-CAT) 22-ID beamline at the Advanced Photon Source, Argonne National Laboratory. Supporting institutions may be found at www.ser-cat.org/members.html. We are grateful to SER-CAT staff for experimental support. Use of the Advanced Photon Source was supported by the U.S. Department of Energy, Office of Science, Office of Basic Energy Sciences, under Contract No. W-31-109-Eng-38. This work was supported in part by ALSAC, grant R01GM098771 (to E.J.E.) from NIGMS, and by Comprehensive Cancer Center Support Grant 5 P30 CA021765 (to St. Jude).

## Author contributions

M.M., L.B.E. and E.J.E. designed and performed the study. All authors contributed to writing the manuscript.

## Additional information

**Competing interests:** The authors declare no competing interests.

