## [Peer Review File · Nature Communications]

Reviewers' comments:

Reviewer #1 (Remarks to the Author):

The manuscript from Enemark and colleagues describe the first high-resolution structure of an archaeal DNA replicative helicase in complex with single stranded DNA and nucleotide cofactors. While there have been prior low resolution electron microscopic structures of homologous helices bound to duplex DNA, this high resolution view allows for the identification of residues that interact with the DNA substrate and points towards a unifying mechanism for DNA translocation. While the results of the study are of significance to warrant publication in Nature Communications, the authors could do better to make the manuscript more amenable for non-specialists that make up the readership of a broadly-read journal such as Nature Comm. I would advocate for publication but only upon addressing the following points:

Major point:

1. The crystallographic studies would be enhanced significantly with some mutational analysis. Even if it done at the level of end-point qualitative analysis, mutations of residues highlighted in Figure 2 would really make for a much stronger story.

Minor points:

1. Suppl Fig 1 really belongs in the main text. These data are crucial for illustrating the viability and justification of the construct used for crystallographic studies. The authors should also include a diagrammatic representation of the constructs tested (i.e. a domain diagram showing WT and the variant used).
2. Line 80: casual readers may not know what the Sc in ScMCM stands for. Please elaborate for clarity.
3. Line 55: technically, the "atoms" of any polypeptide are conserved (they are always either C, N, O, P or S). Please rephrase.

Reviewer #2 (Remarks to the Author):

Meagher and colleagues present the crystal structure of the MCM complex from archaeon *S. solfataricus* bound to single-stranded DNA, and to a combination of ADP, and ATP-mimic, ADP-BeF₃. The atomic model appears reminiscent of the available cryo-EM structure of single-stranded DNA engaged MCM ring, contained in the 11-subunit complex Cdc45-MCM-GINS (CMG) from yeast (indeed I feel a figure comparing the two structures is lacking in the manuscript). The resolution of the crystal structure presented here is significantly improved compared to the yeast cryo-EM map (from 4.9 to 3.1 Å), providing a better description of

1. how the PS1 and H2I hairpins contact single-stranded DNA
2. How ATPase site occupancy influences the orientation of MCM channel hairpins.

The structure also confirms the polarity of DNA translocation previously observed in the yeast CMG (from the Li/O'Donnell and Diffley) and CMG-Pol epsilon complexes (from Costa).

The new crystallographic data is solid and deserves to be published.

Puzzlingly however, the authors claim that their atomic model supports a rotary cycling mechanism for DNA translocation. This is based on molecular morphing between rotated copies of the same crystal structure, which give the perception of rotation-mediated single-stranded DNA translocation. This approach was used before by Enemark and Joshua-Tor in an important paper published in 2006, on the E1 replicative helicase from Papillomavirus. While the earlier work was technically very impressive at the time and provided a useful model that influenced the field of DNA translocases significantly, modern technology and more advanced model systems available today render the current study's approach to modeling translocation obsolete.

In fact, in a 2017 PNAS paper, Huilin Li and Mike O'Donnell have used high-resolution cryo-EM to image ATP-hydrolysis driven translocation on a DNA fork, by the yeast CMG helicase.

In the yeast experiment the CMG was frozen as it was hydrolyzing ATP and imaged, while the present *Sulfolobus* study employs ADP-BeF₃ and ADP to mimic pre and post hydrolysis states.

In the yeast study, ATPase channel hairpins were visualized contacting single-stranded DNA, as well as duplex DNA being split as it enters the MCM channel (a feature missing in the *Sulfolobus* structure). The CMG model is more complex compared to the homomeric archaeal MCM, for example because MCM is made of six different subunits. Working with the yeast heterohexamers allows the actual discrimination of distinct translocation states within the helicase ring, which is impossible with a homohexamer (at least with the construct used in this study).

Importantly, in the translocating yeast CMG, ssDNA is bound to the Mcm2-5-3 ATPase subunits and is not observed to rotate around the ring, raising the question of whether MCM indeed translocates on DNA using a rotary-cycling mechanism at all. An alternative mechanism for translocation has been proposed by the Li and O'Donnell groups, whereby the helicase extends and compresses, as it translocates along the DNA. The current crystal structure of the homomeric MCM cannot discriminate between the two models, as it only shows one of the two states previously observed in yeast. Indeed, available biochemistry and genetics on eukaryotic MCM argues against a rotary-cycling mechanism for the MCM helicase. In fact, according to studies in *Drosophila*, ATPase mutations can be introduced within some of the hetero-hexameric ring subunits, while the CMG still functions as a helicase. This also holds true for *Sulfolobus* MCM, as ATPase mutants can also be doped into the helicase ring, which do not inactivate the DNA unwinding function of the replicative helicase. Likewise, genetics experiments from Davey in yeast indicate that a lethal phenotype can be observed only when DNA-interacting residues located on the PS1 hairpin of Mcm3 are mutated, but on not other MCM subunits. Taken together, this evidence supports a functional asymmetry within the MCM ring that is hard to reconcile with the proposed rotary cycling mechanism.

Given this available knowledge, I find it surprising that the authors push the idea that the DNA-bound form of MCM described in this manuscript explains the mechanism of DNA translocation by MCM.

The authors build an argument to support the notion that rotary cycling must occur not only in the homomeric archaeal MCM, but also in the heteromeric MCM of eukarya. The key data that should support this notion is that DNA-interacting residues in the channel hairpins are conserved in all six eukaryotic MCM subunits. However, two studies (from Costa and from Li) on the eukaryotic MCM double hexamer bound to double stranded DNA showed that channel hairpins interact with both DNA strands, and not only with the translocation strand. Therefore, conservation of DNA interacting residues in all MCM subunits in eukaryotes might only be important for MCM-DNA interactions that precede activation of the DNA unwinding function, and have nothing to do with

DNA translocation.

The discussion section contains a very nice, yet extremely speculative, model for the mechanism of double hexamer separation, duplex DNA melting and ATP-hydrolysis driven DNA replication. I really like this model however I note that it is not supported by any experimental data, and importantly it is incompatible with recent evidence published by John Diffley, who used the reconstituted DNA replication system to show that CMG formation promotes separation of the MCM double hexamer before the MCM ATP hydrolysis function is switched on.

In conclusion, I think the data presented in the results section is solid and it should be described for what it is: an improvement in resolution for a protein-DNA structure from an archaeal model system, which provides important information on the MCM ATPase interaction with DNA. The authors should clearly state that the mechanisms of rotary cycling and origin melting is only speculative.

Reviewer #3 (Remarks to the Author):

The paper by Meagher et al. addresses the mechanism of the DNA helicase MCM from an archaeal species that is highly related in primary and secondary structure to the eukaryotic Mcm2-7 helicase and also distantly related to eukaryotic virus DNA helicases. The current structure shows a near full-length helicase bound to single strand DNA (ssDNA) and is of sufficient resolution that protein-DNA interactions can be understood. The data are consistent with the senior authors' previous work on the papillomavirus E1 protein and with the two-base step mechanism of DNA translocation driven by ATP hydrolysis. Based on this data, the main part of the discussion proposes a very detailed mechanism for how the helicase might transition from the double-strand bound Mcm2-7 double hexamer (DH) to the unwound DNA where two helicases pass each other. The model addresses published data with dsDNA bound eukaryotic Mcm2-7 DH showing that the two DNA lagging strands are bound to the opposite Mcm2-7 subunits, in contrast to the observation in this paper that the ssDNA binding would need to transition to the leading strand ssDNA. They propose a sliding mechanism for the DH, followed by twisting of the hexamers to extrude the ssDNA between the DH interphase.

The model is speculative, but very plausible and is based on existing data and the data presented in the current paper. As such, it is very appropriate that the model be put out for testing.

One additional piece of data is requested. In Supplementary Fig. 1. The results should be quantitated and in addition to the gel shown, also shown as a graph with statistics. It looks as if the truncated protein might be more active, but from the data presented it is not possible to tell.

We thank the reviewers for careful consideration of our manuscript. All reviewers indicated that our suggested mechanism of MCM activation is speculative. We certainly consider this as unproven, perhaps best termed a “hypothesis.” We have put the term “hypothesis” in the section header, which we think appropriately characterizes that this is an idea that is not proven by the present structure or experiments. We are grateful for the reviewers open-minded consideration of the idea and how we have presented it.

Based on the comments of reviewers 1 and 3, we have added a strand displacement mutational analysis for the residues that contact DNA in our structure and have quantified the results of the assays. Overall, we feel that this has made for a much stronger manuscript. The experiments suggest that the crystallized construct has a larger degree of cooperativity than the full-length construct. The helix-2-insert T369A mutant was horribly aggregated for full-length, but it could be purified in the context of the crystallized construct. This region of the structure also became an area of interest in addressing the comments of reviewer 2. Specifically, this portion of the helix-2-insert adopts a “horseshoe” conformation for the Mcm3, Mcm5, and Mcm2 subunits in both of the yeast MCM double-hexamer structures, but these “horseshoe” conformations seriously clash with the pre-sensor-1- β hairpin of the neighboring subunit in the CMG:DNA models (See Fig. 5). We now suggest that conversion of the MCM double-hexamer from encircling double-stranded DNA to binding single-stranded DNA involves the conversion of the helix-2-inserts of the Mcm3, Mcm5, and Mcm2 subunits from “horseshoe” to “helix”, allowing them to engage in binding single-stranded DNA.

Responses to specific points raised by the reviewers are provided below in **yellow highlight**.

Reviewers' comments:

Reviewer #1 (Remarks to the Author):

The manuscript from Enemark and colleagues describe the first high-resolution structure of an archaeal DNA replicative helicase in complex with single stranded DNA and nucleotide cofactors. While there have been prior low resolution electron microscopic structures of homologous helices bound to duplex DNA, this high resolution view allows for the identification of residues that interact with the DNA substrate and points towards a unifying mechanism for DNA translocation. While the results of the study are of significance to warrant publication in Nature Communications, the authors could do better to make the manuscript more amenable for non-specialists that make up the readership of a broadly-read journal such as Nature Comm. I would advocate for publication but only upon addressing the following points:

Major point:

1. The crystallographic studies would be enhanced significantly with some mutational analysis. Even if it done at the level of end-point qualitative analysis, mutations of residues highlighted in Figure 2 would really make for a much stronger story.

We have added a mutational analysis of residues that we identified from the structure to contact ssDNA (K430, T369, and Y386). This analysis has been performed for the full-length SsoMCM protein and also for the construct of the study (SsoMCM-GGSGGS- Δ C). We have provided the parameters of activity as a main table (Table 2). The gels, quantification, and parameter fitting for these experiments are illustrated in Supplementary Figure 4. We thank the reviewer for the suggestion and feel that these have significantly improved the story. We note that the full-length SsoMCM-K430A mutant has previously been shown to be inactive for strand displacement (McGeoch et al, NSMB, 2005), and the K430A mutants of our

experiments show similar inactivity. We could not test the T369A mutant of full-length SsoMCM in the assay because this protein was very badly aggregated and irreversibly clogged our size-exclusion column. The T369 residue is on a region prone to variable structure. In addition to the "horseshoe" and "helix" conformations our present manuscript describes, we have previously shown the capacity of the helix-2-insert to adopt two distinctly different strand topologies—one forming a hairpin separate from the pre-sensor-1- β -hairpin (as in the structure described here), and one forming an extended β -sheet with the pre-sensor-1- β -hairpin as an interface between two octamers of a double-octamer (Miller et al, eLife, 2014). We therefore suggest that the T369A mutation may disrupt structure for the full-length SsoMCM protein, but that the greater structural constraints of the shorter interdomain linker of SsoMCM-GGSGGS- Δ C sufficiently reduces that disruption to allow isolation a soluble protein sample.

Minor points:

1. Suppl Fig 1 really belongs in the main text. These data are crucial for illustrating the viability and justification of the construct used for crystallographic studies. The authors should also include a diagrammatic representation of the constructs tested (i.e. a domain diagram showing WT and the variant used).

We have placed 2 gel images as Figure 1 (full-length SsoMCM and SsoMCM-GGSGGS- Δ C) and added a domain diagram. We note that uncropped versions of the same 2 gel images are provided within Supplementary Figure 4 to assist comparison in the mutational analysis.

2. Line 80: casual readers may not know what the Sc in ScMCM stands for. Please elaborate for clarity.

We have defined the abbreviation at line 80.

3. Line 55: technically, the "atoms" of any polypeptide are conserved (they are always either C, N, O, P or S). Please rephrase.

We have changed the wording at line 56 to say that the identified contacts are fully possible for all MCM proteins. Two of the interactions that we describe involve the backbone amides of the polypeptide (A431 and V377), which would be generally possible for any amino acid if not proline. This aspect is described in the legend of Supplementary Figure 3. Another interaction involves the hydroxyl group of a side-chain that is conserved as serine or threonine (T369). The last interaction is an ionic interaction (K430) involving a residue that is lysine at every sequence we show in the alignment (Supplementary Fig 3) except *DmMcm6* (where it is arginine).

Reviewer #2 (Remarks to the Author):

Meagher and colleagues present the crystal structure of the MCM complex from archaeon *S. solfataricus* bound to single-stranded DNA, and to a combination of ADP, and ATP-mimic, ADP-BeF₃. The atomic model appears reminiscent of the available cryo-EM structure of single-stranded DNA engaged MCM ring, contained in the 11-subunit complex Cdc45-MCM-GINS (CMG) from yeast (indeed I feel a figure comparing the two structures is lacking in the manuscript). The resolution of the crystal structure presented here is significantly improved compared to the yeast cryo-EM map (from 4.9 to 3.1 Å), providing a better description of

1. how the PS1 and H2I hairpins contact single-stranded DNA

2. How ATPase site occupancy influences the orientation of MCM channel hairpins.

The structure also confirms the polarity of DNA translocation previously observed in the yeast CMG (from the Li/O'Donnell and Diffley) and CMG-Pol epsilon complexes (from Costa).

The new crystallographic data is solid and deserves to be published.

Puzzlingly however, the authors claim that their atomic model supports a rotary cycling mechanism for DNA translocation. This is based on molecular morphing between rotated copies of the same crystal structure, which give the perception of rotation-mediated single-stranded DNA translocation. This approach was used before by Enemark and Joshua-Tor in an important paper published in 2006, on the E1 replicative helicase from Papillomavirus. While the earlier work was technically very impressive at the time and provided a useful model that influenced the field of DNA translocases significantly, modern technology and more advanced model systems available today render the current study's approach to modeling translocation obsolete.

In fact, in a 2017 PNAS paper, Huilin Li and Mike O'Donnell have used high-resolution cryo-EM to image ATP-hydrolysis driven translocation on a DNA fork, by the yeast CMG helicase.

In the yeast experiment the CMG was frozen as it was hydrolyzing ATP and imaged, while the present *Sulfolobus* study employs ADP-BeF3 and ADP to mimic pre and post hydrolysis states.

Our structure shows clear differences at the ATPase sites around the ring (Fig. 3 and Supplementary Movie 1) that are consistent with the structural events of the ATP hydrolysis cycle. The crucial factor is that our structure provides sufficient resolution to discriminate these states, which are all very similar to one another. The resolution of the yeast CMG structure does not permit visualization of such subtle differences. The "all-or-none" assignments in the yeast CMG structure with ATP at 3 of the sites and empty at the other 3 sites would suggest that hydrolysis products (such as ADP) are not present at any of the ATPase sites, which would make it inaccurate to describe this as "frozen as it was hydrolyzing" because all the species described are at the "reactant" side of the hydrolysis reaction, and none are at the "product" side. The assigned nucleotide configurations probably make the structure more consistent with "frozen prior to hydrolyzing" than "frozen as it was hydrolyzing". A specific description of the ATPase sites of the yeast CMG structures as "hydrolyzing" or "pre-hydrolyzing" is ultimately speculative because the CMG:DNA structures do not have adequate resolution to differentiate these scenarios.

In the yeast study, ATPase channel hairpins were visualized contacting single-stranded DNA, as well as duplex DNA being split as it enters the MCM channel (a feature missing in the *Sulfolobus* structure). The CMG model is more complex compared to the homomeric archaeal MCM, for example because MCM is made of six different subunits. Working with the yeast heterohexamers allows the actual discrimination of distinct translocation states within the helicase ring, which is impossible with a homohexamer (at least with the construct used in this study).

Our structure clearly demonstrates six discrete positions for the hairpins of the six subunits in a “staircase.” This arrangement removes the 6-fold symmetry that might exist in a homohexamer in the absence of DNA, such as in the highly symmetric DNA-free MCM hexamer structure that we described in eLife (Miller et al, eLife, 2014; PDB: 4R7Y). With a 6-fold symmetric species, all 6 ring permutations superimpose perfectly, and the 6 permutations all represent a single (symmetric) state. If the symmetry is broken (as when engaged in a ssDNA-binding staircase), the 6 permutations no longer superimpose on each other and thus represent 6 distinct states. The identical chemical composition of the subunits in a homomeric hexamer causes these 6 states to be energetically equivalent. In short, a “staircased” hexamer can be constructed as ABCDEF, BCDEFA, CDEFAB, DEFABC, EFABCD, FABCDE with the subunit order indicating relative “height” in the staircase. All of these 6 permutations are energetically identical because the constituent subunits are chemically identical. As such, the structure of any one permutation precisely provides the structure of all 6 equivalent states by passing the ring through the 6 permutations. Thus, our structure of one single asymmetric hexamer defines 6 states. Analogously, the 2 different hexamers of papillomavirus E1 bound to DNA (Enemark & Joshua-Tor, Nature, 2006; PDB 2GXA) defines 12 states. For a heterohexamer such as eukaryotic Mcm2-7, the ring is intrinsically asymmetric due to the different chemical composition of each subunit. Permutation of the “states” of the ring leads to species that are energetically distinct. For example, a 352647 staircase would differ energetically from 526473. As a result, a single heterohexameric hexamer structure defines one single state. Our manuscript suggests that each subunit occupies the “top” position of the staircase in turn during the translocation cycle, indicating that at least 5 eukaryotic CMG states are yet to be observed structurally because reported CMG:DNA structures have exclusively shown Mcm3 at the “top” position. We have generally kept the issue of discrete “states” at the level of describing the different hairpin “heights” within a staircase and that hairpins ascend to the “top” of the staircase upon ATP-binding and that the collective staircase descends associated with one ATP-hydrolysis event.

For homo-hexameric MCM species, the subtle differences at the ATPase sites cannot be attributed to chemical differences among the subunits themselves, and it is mathematically straightforward to align and view all the ATPase sites from a consistent perspective (as in Fig. 3 and Supplementary Movie 1). Consequently, the subtle differences observed likely reflect the ATPase cycle itself as opposed to subunit differences.

Importantly, in the translocating yeast CMG, ssDNA is bound to the Mcm2-5-3 ATPase subunits and is not observed to rotate around the ring, raising the question of whether MCM indeed translocates on DNA using a rotary-cycling mechanism at all. An alternative mechanism for translocation has been proposed by the Li and O’Donnell groups, whereby the helicase extends and compresses, as it translocates along the DNA. The current crystal structure of the homomeric MCM cannot discriminate between the two models, as it only shows one of the two states previously observed in yeast. Indeed, available biochemistry and genetics on eukaryotic MCM argues against a rotary-cycling mechanism for the MCM helicase. In fact, according to studies in *Drosophila*, ATPase mutations can be introduced within some of the heterohexameric ring subunits, while the CMG still functions as a helicase. This also holds true for *Sulfolobus* MCM, as ATPase mutants can also be doped into the helicase ring, which do not inactivate the DNA unwinding function of the replicative helicase. Likewise, genetics experiments from Davey in yeast indicate that a lethal phenotype can be observed only when DNA-interacting residues located on the PS1 hairpin of Mcm3 are mutated, but on not other MCM subunits. Taken together, this evidence supports a functional asymmetry within the MCM ring that is hard to reconcile with the proposed rotary cycling mechanism.

The yeast CMG:DNA structure illustrates one discrete state. The translocation of DNA involves more than one single state. As such, other CMG:DNA species must exist that were not described in the CMG:DNA structure report. The six states defined by our structure illustrate how some of the as-yet unobserved CMG:DNA states likely look. The earlier Li and O’Donnell structures of CMG (NSMB, 2016) were very important in illustrating the arrangement of Cdc45 and GINS with respect to Mcm2-7-- with the highest resolution to date, but these structures lacked DNA and also lacked nucleotide co-factors. They also placed one of the MCM HTH domains inside the central channel where the DNA would be during translocation. Discussion of translocation mechanism intrinsically relies on both DNA and the ATPase

cycle, and we therefore don't consider the NSMB structures of CMG necessarily relevant to discussing translocation mechanism. We have used these structures in placing Cdc45 and GINS with respect to the MCM ring in our activation hypothesis and have cited them in this. Similarly, we do not use our own structure of the *SSo-Pf*MCM hexamer to discuss DNA translocation mechanism because that structure also lacks DNA and lacks ATP with its exclusively ADP-bound nucleotides.

We don't believe that the behavior of ATPase and DNA-binding mutants can be predicted so precisely in establishing or ruling out a rotary mechanism. This has been a common question over the years for the E1 helicase also. The premise of the question is that if all subunits participate in ATP hydrolysis and DNA-binding, then ablating any one of them should completely shut down the entire machine. This is not necessarily the case. As an analogy, a 6-cylinder car engine uses all of its available cylinders for combustion and follows a specific ignition timing sequence among the cylinders. However, if one cylinder fails to fire, the car can still drive. The question, then, becomes what to expect upon removing one, two, or more spark plugs from the engine. We can predict that if all are removed, then the engine will not work; and if all are functional, then the engine should run. We don't have a good *a priori* expectation for the outcome when an intermediate number are removed. We feel the outcome would vary depending on the relative arrangement of disabled cylinders within the ignition sequence, but that the outcome is ultimately uncertain. As another analogy, a bicycle chain drive can serve as a rotary drive model similar to the mechanism described here with the chain functioning as the DNA and the gear teeth as the DNA-binding elements. Similar to above, although all teeth of the gear generally participate in driving the chain, the removal of one tooth likely will not completely disable the bicycle. We envision that the relative effect of removing a specific tooth might be associated with its systematic position in a timing sequence (if any—such as systematically at the start position following a gear change), but the effect is ultimately uncertain.

Regarding Mcm3 hairpin studies, we note that our proposed activation mechanism places special importance on Mcm3 because both steps of significant unwinding occur with the Mcm3 subunit moving to the top of the staircase.

Given this available knowledge, I find it surprising that the authors push the idea that the DNA-bound form of MCM described in this manuscript explains the mechanism of DNA translocation by MCM.

The authors build an argument to support the notion that rotary cycling must occur not only in the homomeric archaeal MCM, but also in the heteromeric MCM of eukarya. The key data that should support this notion is that DNA-interacting residues in the channel hairpins are conserved in all six eukaryotic MCM subunits. However, two studies (from Costa and from Li) on the eukaryotic MCM double hexamer bound to double stranded DNA showed that channel hairpins interact with both DNA strands, and not only with the translocation strand. Therefore, conservation of DNA interacting residues in all MCM subunits in eukaryotes might only be important for MCM-DNA interactions that precede activation of the DNA unwinding function, and have nothing to do with DNA translocation.

We feel that the “null hypothesis” has to be that all MCM rings (homomeric or heteromeric) translocate DNA fundamentally by the same mechanism. We now provide additional support that the conserved residues that bind DNA in our structure also interact with DNA in eukaryotic CMG by illustrating that the identified interactions are structurally reasonable within the yeast CMG:DNA EM structure. This analysis identified that the helix-2-inserts of Mcm3, Mcm5, and Mcm2 of the deposited CMG:DNA models (PDB 5U8S and 5U8T) seriously clash with the pre-sensor-1- β of the neighboring subunit (see Fig. 5). The clashing region of the model is a “horseshoe” conformation of the helix-2-insert, which differs significantly from the helical conformations in the helix-2-inserts of our present structure. These clashes are removed by converting the CMG model at Mcm3, Mcm5, and Mcm2 from “horseshoe” conformations to the “helix” conformations of our present structure. Notably, these conformational changes are required specifically at the 3 subunits assigned to interact with DNA in the yeast CMG:DNA structure. The clash and the conformational change needed to remove it is similar for all three subunits. We specifically illustrate the

clash and change in Fig. 5 and in Supplementary Movie 3 for one interface-- the Mcm5 helix-2-insert and the Mcm2 pre-sensor-1- β . We note that "horseshoe" conformations in Mcm3, Mcm5, and Mcm2 are present in both the DNA-bound and the DNA-free double hexamer structures of Mcm2-7 (PDB 3JA8 and 5BK4) and that the "horseshoe" is well-defined in the corresponding EM maps (EMD_6338 and EMD_9400). Also, the helix-2-insert of the DNA-free hexamer of Sso-PfMCM (PDB 4R7Y) is in a "horseshoe" conformation (not illustrated in the present manuscript). After remodeling the helix-2-insert, the yeast CMG:DNA structure readily conforms to the interactions identified in our structure with a fit to the deposited density (EMD_8518) that appears comparable to that of the original model.

The discussion section contains a very nice, yet extremely speculative, model for the mechanism of double hexamer separation, duplex DNA melting and ATP-hydrolysis driven DNA replication. I really like this model however I note that it is not supported by any experimental data, and importantly it is incompatible with recent evidence published by John Diffley, who used the reconstituted DNA replication system to show that CMG formation promotes separation of the MCM double hexamer before the MCM ATP hydrolysis function is switched on.

We greatly thank the reviewer for bringing this to our attention. We have made it a strong priority to illustrate consistency of our proposed mechanism of activation with the manuscript of the Diffley lab (Douglas et al, Nature, 2018). Indeed, the calculated unwinding plots of our manuscript were generated for this specific purpose. In striving to illustrate consistency of the unwinding steps, we did not explicitly reconcile ATP hydrolysis timing. We now refer to the Diffley manuscript on the issue of nucleotide removal and also that ATP hydrolysis initiates after recruitment of Mcm10. We now explicitly describe Mcm10 recruitment to precede the series of ATP hydrolysis events (steps 3-7 of Fig. 8) prior to final strand exit. We also explicitly state that the energetic driving force for steps 1 and 2 of Fig. 8 derives from the binding of protein factors (Cdc45 and GINS) and not hydrolysis of ATP.

In conclusion, I think the data presented in the results section is solid and it should be described for what it is: an improvement in resolution for a protein-DNA structure from an archaeal model system, which provides important information on the MCM ATPase interaction with DNA. The authors should clearly state that the mechanisms of rotary cycling and origin melting is only speculative.

Reviewer #3 (Remarks to the Author):

The paper by Meagher et al. addresses the mechanism of the DNA helicase MCM from an archaeal species that is highly related in primary and secondary structure to the eukaryotic Mcm2-7 helicase and also distantly related to eukaryotic virus DNA helicases. The current structure shows a near full-length helicase bound to single strand DNA (ssDNA) and is of sufficient resolution that protein-DNA interactions can be understood. The data are consistent with the senior authors previous work on the papillomavirus E1 protein and with the two-base step mechanism of DNA translocation driven by ATP hydrolysis. Based on this data, the main part of the discussion proposes a very detailed mechanism for how the helicase might transition from the double strand bound Mcm2-7 double hexamer (DH) to the unwound DNA where two helicases pass each other. The model addresses published data with dsDNA bound eukaryotic Mcm2-7 DH showing that the two DNA lagging strands are bound to the opposite Mcm2-7 subunits, in contrast to the observation in this paper that the ssDNA binding would need to transition to the leading strand ssDNA. They propose a sliding mechanism for the DH, followed by twisting of the hexamers to extrude the ssDNA between the DH interphase.

The model is speculative, but very plausible and is based on existing data and the data presented in the current paper. As such, it is very appropriate that the model be put out for testing.

One additional piece of data is requested. In Supplementary Fig. 1. The results should be quantitated and in addition to the gel shown, also shown as a graph with statistics. It looks as if the truncated protein might be more active, but from the data presented it is not possible to tell.

We have quantified strand displacement by the crystallized construct (SsoMCM-GGSGGS- Δ C), full-length SsoMCM, and the single mutants K430A, T369A, and Y386A of these. As described above, full-length SsoMCM T369A was badly aggregated and could not be tested in the assay. The activity parameters for the 7 proteins tested are provided in Table 2 of the main text. The gels, graphical quantitation, and parameter fits are illustrated in Supplementary Fig. 4. Indeed, the truncated protein seems to have more cooperativity than full-length. This trend appears present for wild-type and also the Y386A mutant. The T369A mutant arguably showed the strongest difference for full-length versus truncated in that full-length was horribly aggregated while the truncated version could be purified and was active—though it appeared to be the least cooperative of the proteins that could be tested.

REVIEWERS' COMMENTS:

Reviewer #1 (Remarks to the Author):

The authors have addressed each of the comments raised during the initial review of the manuscript. I think for the non-specialists, the authors should include some statements justifying their choice for data resolution cutoff. To the non-specialist, the high merging R value and low signal to noise in the highest resolution shell may be off-putting. The authors should assuage any concerns by explaining their rationale (citing the appropriate references as needed).

Reviewer #2 (Remarks to the Author):

The authors carefully addressed most of the points I raised and also include DNA unwinding assays using structure-based mutants of MCM that target DNA-interacting residues. The new data definitely strengthen this study.

The authors have toned down many of their speculative claims, for example they now refer to their model for replication initiation as a hypothesis, which I feel is fair. I still feel certain sentences should be reworked.

One sentence in the abstract should be removed, as insufficient evidence is provided to back up the claim. In the reply to my comments, the authors highlight that "for a heterohexameric such as [...] Mcm2-7, the ring is intrinsically asymmetric due to different chemical composition of each subunit. Permutation of the "states" of the ring leads to species that are energetically distinct [...]" The authors identify residues that contact DNA in the *Sulfolobus* MCM structure and are conserved in higher eukaryotes. However, the authors do not show that different rotational states exist within the Mcm2-7 ring. Because of these facts, I do not think that enough evidence is provided for the statement "This rotary mechanism appears conserved for all eukaryotes [...]".

I also have a problem with the phrasing of the following sentence on page 2. "The structure indicates that the MCM complex uses a "rotary" mechanism to translocate on DNA". I think this statement is misleading. In their reply to my comments, the authors provide a nice explanation of their rationale for how rotary cycling can be modeled by propagating the same asymmetric DNA binding state around a homo-hexameric ring. I feel including this explanation can be very helpful and can lead the reader to accept the proposed model.

Finally, I feel the authors should acknowledge, at least when discussing eukaryotes, that the CMG helicase can tolerate inactivating mutations. Botchan and colleagues have shown that an Mcm6 Walker A mutant contains near wild type DNA unwinding activity, while Mcm5 and Mcm3 Walker A mutants do not support unwinding. What this means in terms of the DNA translocation mechanism is unclear (the authors provide a useful discussion in the reply to my comments), but the data ATPase mutagenesis data should be acknowledged when presenting the mechanism.

Reviewer #3 (Remarks to the Author):

The authors have revised the manuscript by adding new data on the analysis of wild type and mutant versions of the helicase and have quantified the results in Table 2. They have also appropriately responded to the reviewers comments about the model. The paper now represents an important contribution to the field because of the increased resolution of the structure and the single stranded DNA present in the structure. The model is entirely appropriate to suggest based on the data in this paper and others.

Bruce Stillman

We thank the reviewers for careful consideration of our revised manuscript. Responses to specific points raised by the reviewers are provided below in yellow highlight.

REVIEWERS' COMMENTS:

Reviewer #1 (Remarks to the Author):

The authors have addressed each of the comments raised during the initial review of the manuscript. I think for the non-specialists, the authors should include some statements justifying their choice for data resolution cutoff. To the non-specialist, the high merging R value and low signal to noise in the highest resolution shell may be off-putting. The authors should assuage any concerns by explaining their rationale (citing the appropriate references as needed).

This is an important point that deserves the attention suggested by the reviewer. We thank the reviewer for noting this. The data exhibit a signal-to-noise greater than 1.0 up to 3.3 Å resolution. Beyond 3.3 Å, the signal-to-noise decreases below 1.0. Based on two publications (now cited as references 54 and 55), we feel it is appropriate to include all data to 3.15 Å resolution during crystallographic refinement because this is where the CC1/2 value falls to 0.245. This represents a cutoff that exceeds the 0.1 value described in reference 54 and falls within the 0.2-0.4 range suggested by reference 55 as the range where little information remains in the diffraction data.

We feel that the resolution cutoff number matters less than the appearance of the electron density maps in the specific regions of interest. In considering ssDNA translocation mechanism, the critical features are the ssDNA and the ATPase sites. Accordingly, we have strived to illustrate the electron density for the DNA (Supplementary Fig. 1a) and the ATPase sites (Fig. 3) prior to inclusion of ssDNA or BeF₃, respectively, in the model— so that the specific density illustrated is independent of model bias. These specific maps are probably the best fundamental indicators of how finely structural details can be discriminated in these particular regions. We also show maps of these regions derived from the final crystallographic refinement (Supplementary Fig. 1b and Supplementary Movie 1). These maps are generally less noisy and can show greater detail than the “unbiased” maps generated earlier in the model-building/refinement process. However, the possibility of model bias ultimately can never be excluded for maps derived from any stage of refinement that is subsequent to the incorporation of the specific attribute of interest to the model. Collectively, the two types of maps (one: prior to incorporating the attribute of interest in the model; and two: following the final refinement) complement one another, and both are provided to maximize the reader’s ability to assess the regions of focus in our structure.

Reviewer #2 (Remarks to the Author):

The authors carefully addressed most of the points I raised and also include DNA unwinding assays using structure-based mutants of MCM that target DNA-interacting

residues. The new data definitely strengthen this study.

The authors have toned down many of their speculative claims, for example they now refer to their model for replication initiation as a hypothesis, which I feel is fair. I still feel certain sentences should be reworked.

One sentence in the abstract should be removed, as insufficient evidence is provided to back up the claim. In the reply to my comments, the authors highlight that “for a heterohexameric such as [...] Mcm2-7, the ring is intrinsically asymmetric due to different chemical composition of each subunit. Permutation of the “states” of the ring leads to species that are energetically distinct [...]” The authors identify residues that contact DNA in the *Sulfolobus* MCM structure and are conserved in higher eukaryotes. However, the authors do not show that different rotational states exist within the Mcm2-7 ring. Because of these facts, I do not think that enough evidence is provided for the statement “This rotary mechanism appears conserved for all eukaryotes [...]”.

Yes— we now see that the word “appear” could have different implications. To us, the sequence conservation made the mechanism “appear” conserved, but this could also be interpreted to mean a visible appearance (an observation) of a eukaryotic Mcm2-7 complex-- which our manuscript does not have. We have reworked the sentence to clearly state that its basis is sequence conservation: “Sequence conservation indicates this rotary mechanism is fully possible for all eukaryotes and archaea”. Following this change, a few minor modifications were needed to shorten the overall abstract to meet the journal guidelines.

I also have a problem with the phrasing of the following sentence on page 2. “The structure indicates that the MCM complex uses a “rotary” mechanism to translocate on DNA”. I think this statement is misleading. In their reply to my comments, the authors provide a nice explanation of their rationale for how rotary cycling can be modeled by propagating the same asymmetric DNA binding state around a homo-hexameric ring. I feel including this explanation can be very helpful and can lead the reader to accept the proposed model.

We now provide additional details for the use of the term “rotary catalysis” based on the original description for F1-ATPase (added Ref 46) and also cite prior formalism of hexameric ring helicases based on the F1-ATPase rotary catalysis model (Ref 1). In each case, ATP hydrolysis proceeds sequentially around the ring (as depicted in Fig. 4 and Supplementary Movie 2), and hence the specific ATPase site that is catalytically active (T^*) rotates around the ring. We also note that a rotary model would unify all super families of hexameric ring helicases and peptide translocases also use a rotary translocation mechanism.

Finally, I feel the authors should acknowledge, at least when discussing eukaryotes, that the CMG helicase can tolerate inactivating mutations. Botchan and colleagues have shown that an Mcm6 Walker A mutant contains near wild type DNA unwinding activity, while Mcm5 and Mcm3 Walker A mutants do not support unwinding. What this means in

terms of the DNA translocation mechanism is unclear (the authors provide a useful discussion in the reply to my comments), but the data ATPase mutagenesis data should be acknowledged when presenting the mechanism.

We have added a brief discussion of DNA unwinding by Walker-A ATPase site mutants of CMG reported in the Botchan manuscript (Ref 24). That manuscript illustrates that lysine to alanine Walker-A mutants of Mcm3 and Mcm5 are highly defective in unwinding DNA. The ATPase sites affected by these mutations are at the Mcm3/5 and the Mcm5/2 subunit interfaces. Notably, the subunits of these interfaces (Mcm3, 5, and 2) are the same three subunits where the helix-2-insert remodels from horseshoe to helix to bind ssDNA (see Fig. 5). This remodeling is associated with bringing adjacent subunits closer together (see Supplementary Movie 3)— which is precisely what occurs upon ATP-binding. As such, it is highly plausible that the underlying reason that the Walker-A mutants of Mcm3 and Mcm5 are defective in unwinding DNA is that they fail to tighten their intersubunit interfaces sufficiently to drive the helix-2-inserts of Mcm3, 5, and 2 from horseshoe to helix. Without such remodeling, the helix-2-inserts of these subunits would be in conformations more similar to those of the inert double-hexamer than the unwinding-competent form depicted in our structure. While not certain, such an effect might have more to do with defects in ATP binding than in ATP hydrolysis. Additionally, although a rotary mechanism calls for ATP hydrolysis at each ATPase site in general, the ablation of one site (removal of one spark plug) does not necessarily disable the engine.

Reviewer #3 (Remarks to the Author):

The authors have revised the manuscript by adding new data on the analysis of wild type and mutant versions of the helicase and have quantified the results in Table 2. They have also appropriately responded to the reviewers comments about the model. The paper now represents an important contribution to the field because of the increased resolution of the structure and the single stranded DNA present in the structure. The model is entirely appropriate to suggest based on the data in this paper and others.

Bruce Stillman